# Attention-to-Survival: Multimodal Fracture Risk Prediction Based on Pelvic Radiographs and Clinical Data from the Study of Osteoporotic Fractures

**Niklas C. Koser**[*1] [iD]                NIKLAS.KOSER@RAD.UNI-KIEL.DE
**Marten J. Finck**[*2] [iD]                MAFI@INFORMATIK.UNI-KIEL.DE
**Silja Janßen**[3] [iD]                SJA@INFORMATIK.UNI-KIEL.DE
**Coenraad Mouton**[1] [iD]                COENRAAD.MOUTON@RAD.UNI-KIEL.DE
**Li-Y. Lui**[4]                LILY.LUI@UCSF.EDU
**Steven R. Cummings**[4]                STEVEN.CUMMINGS@UCSF.EDU
**Kevin Köser**[3] [iD]                KK@INFORMATIK.UNI-KIEL.DE
**Jan-B. Hövener**[1] [iD]                JAN.HOEVENER@RAD.UNI-KIEL.DE
**Sören Pirk**[2] [iD]                SP@INFORMATIK.UNI-KIEL.DE
**Claus-C. Glüer**[1] [iD]                GLUEER@RAD.UNI-KIEL.DE

[1] *i2Lab@SBMI, Kiel University, University Hospital Schleswig-Holstein, Kiel, Germany*

[2] *Visual Computing and Artificial Intelligence, Kiel University, Kiel, Germany*

[3] *Department of Computer Science, Kiel University, Kiel, Germany*

[4] *California Pacific Medical Center Research Institute, San Francisco, CA, USA*

**Editors:** Accepted for publication at MIDL 2026

## Abstract

Osteoporotic changes in the hip structure render the proximal femur particularly vulnerable to fractures, which leads to severe consequences for patients' health and significant socioeconomic burdens, a strongly increasing problem in aging populations. Accurate risk estimation is therefore essential for initiating timely preventive measures. However, the current clinical standard measures bone mineral density (BMD) and the Fracture Risk Assessment Tool (FRAX®) provide only limited predictive value. Neither BMD nor FRAX® capture structural characteristics that could be derived from pelvic radiographs that are widely available. To address this gap, we present the Attention-to-Survival Fusion (ATSF) model, a multimodal survival analysis framework that combines clinical risk factors (CRFs) with pelvic radiograph features. An attention-based architecture equipped with a deep conditional transformation model (DCTM) prediction head enables accurate estimation of time-dependent fracture risk. The ATSF model is designed to accommodate missing clinical variables, handle all forms of non-informative censoring, and provides modality-specific interpretability through the attention mechanisms. It was developed, validated and tested with data of 7825 women from the Study of Osteoporotic Fractures (SOF) followed for fracture incidence for 23 years. We benchmark ATSF against established baselines, including FRAX®, the Cox proportional hazards model (CoxPH), and a deep learning reference model. Our results demonstrate significant superior performance across concordance index (C-index) and area under the receiver operating characteristic curve (AUC), indicating the importance of integrating radiographic and clinical data within a unified survival framework. Furthermore, offering improved interpretability and a scalable multimodal design, the proposed method provides a promising alternative for advancing individualized hip-fracture risk prediction in osteoporosis research and precision medicine.

**Keywords:** Survival Analysis, Hip Fracture Risk, Multimodal, Deep Learning

---

[*] Contributed equally

## 1. Introduction

Osteoporotic fractures resulting from gradual deterioration of the trabecular bone structure represent a major health problem for the elderly population due to their prevalence rising steadily with age (Compston et al., 2019; Maquer et al., 2015; Shen et al., 2022). Usually progressing unnoticed, even minor trauma or everyday stresses can lead to fractures due to the fragility of bone (Bartl, 2010). The resulting increase in morbidity and mortality not only significantly reduces the quality of life of those affected, but also places a considerable socioeconomic burden on healthcare systems worldwide (Shen et al., 2022; Kanis et al., 2021). In order to prevent such consequences, accurate prognosis and early diagnosis are crucial, as timely, guideline-based preventive measures have been proven successful (Shepstone et al., 2018; Porter and Varacallo, 2025).

In the clinical assessment of osteoporosis, the World Health Organization (WHO) recommends determining bone mineral density (BMD) using dual-energy X-ray absorptiometry (DXA) as the clinical standard. The measurement provides a quantitative indicator of bone mineralization and thus offers insights into the extent of bone loss and the risk of future fractures, including the most important fracture sites, vertebral bodies and the proximal femur (Kanis et al., 2007; Bartl, 2010).

Since osteoporosis cannot be characterized solely by decreasing bone density, additional diagnostic metrics, specifically those reflecting bone microarchitecture, have been proposed in the literature to improve the sensitivity of risk assessment. One example is the trabecular bone score (TBS) enabling indirect quantification of trabecular structure (Harvey et al., 2015; Silva et al., 2014). In addition, current research is investigating the use of high-resolution peripheral quantitative computed tomography (HRpQCT) but this method cannot be applied to the hip. To address this, deep learning (DL)-based super-resolution methods of clinical CTs have been developed to derive trabecular structure parameters (Nishiyama and Shane, 2013; Koser et al., 2025; Finck et al., 2025). Aside from image derived metrics clinical risk factors (CRFs) constitute to the patient-specific risk of fracture. These include, among others, gender, age, body mass index, prior fractures, alcohol, and tobacco consumption. Importantly, some of these risk factors are at least partially dependent on BMD, whereas others influence fracture risk independently of BMD (Kanis et al., 2007; Bartl, 2010; Porter and Varacallo, 2025).

Based on these findings, risk models have been developed in recent years that take into account a variety of skeletal and clinical factors in order to predict the absolute risk of fracture within a defined period (usually ten years). A prominent example of this is the Fracture Risk Assessment Tool (FRAX®), which estimates the 10-year risk of hip or major osteoporotic fractures based on CRFs with or without the inclusion of BMD. FRAX® was developed for postmenopausal women and men aged 40 and older and is adapted to the geographical heterogeneity of fracture risk by using country-specific calibrated models. The underlying hazard function is estimated using Poisson regression, a method typically used in survival analysis (SA) (Kanis et al., 2007).

In contrast to the purely binary consideration of the occurrence of an event (e.g., an osteoporotic fracture), SA, also known as time-to-event analysis, allows for a model-based consideration of the temporal dimension of risk. This is particularly important in a clinical context, as both the probability of occurrence and the underlying risk factors can

change over time (Collett, 2023; George et al., 2014; Kanis et al., 2007; Wiegrebe et al., 2024). In addition to conventional methods, DL models for SA have become increasingly important in recent years. These approaches build on the established methods but enable flexible modeling of complex, particularly nonlinear relationships, and allow seamless integration of heterogeneous, multimodal data sources. This provides significantly increased modeling flexibility and can potentially contribute to improved performance in clinical applications (Wiegrebe et al., 2024; George et al., 2014). Further theoretical principles of SA are presented in detail in the appendix A.

The combination of CRFs with image-based features from radiographs represents a promising strategy to improve the prognostic accuracy of survival models for osteoporotic fractures. Prior multimodal approaches in fracture risk prediction remain limited in several respects. For example, Schmarje et al. (2022) classify fracture risk without modeling the temporal component. Similarly, Shaikh et al. (2024) perform convolutional neural network (CNN) based classification of vertebral CT scans to estimate 10-year fracture risk, and subsequently incorporate the CNN-derived prediction into a Cox proportional hazards (CoxPH) model for survival analysis. In contrast, Kong et al. (2022), while conceptually closest to our approach, perform SA on spinal radiographs, incorporating the time component in one multimodal framework. However, they employ a simple ResNet-based architecture trained on a small, single-center dataset. Both Shaikh et al. (2024) and Kong et al. (2022) omit relevant FRAX® CRFs and inherit the proportional hazards limitations of the CoxPH model.

Collectively, the constraints of these methods highlight the need for a unified multimodal survival framework that integrates comprehensive CRFs with pelvic radiographs and directly models the time-dependent fracture risk. Therefore, we aim to advance the state of the art by making the following key contributions: (1) We introduce the Attention-to-Survival Fusion (ATSF) model, an attention-based multimodal network with a deep conditional transformation model (DCTM) prediction head that integrates CRFs with pelvic radiograph features for hip-fracture risk estimation; (2) The model robustly handles missing data and accommodates all forms of non-informative censoring; (3) The attention mechanism provides a pathway for model interpretability and enables insight into modality-specific feature relevance; and (4) We systematically compare our approach with established baselines, including CoxPH (Cox, 1972), FRAX® (Kanis et al., 2007), and the DL approach mentioned (Kong et al., 2022), using concordance index (C-index) as the primary and area under the receiver operating characteristic curve (AUC) as a secondary, complementary metric.

## 2. Related Work

A current and powerful development in the field of DL-based SA are DCTMs (Campanella et al., 2025). This approach parameterizes the log cumulative baseline hazards using Bernstein polynomials and thus also includes the CoxPH model as a special case. The transformation function is modeled by a neural network, which allows complex, nonlinear, and non-proportional hazard structures to be mapped. The architecture consists of a feature extractor for structured or unstructured data and a flexible DCTM survival head. This modular design allows seamless integration with a variety of feature extraction models.

The survival head is trained using the negative log-likelihood (NLL), and its parameterization via Bernstein polynomials and sigmoid distribution functions provides more stable optimization compared to models like DeepSurv, while also avoiding the biases that can arise from cross-entropy-based loss functions, as in DeepHit (Lee et al., 2018). In extensive evaluations DCTMs consistently show performance gains over state-of-the-art (SOTA) models such as DeepSurv (Katzman et al., 2018), DeepHit, random survival forest, and CoxPH (Cox, 1972).

Beyond methodological advances, multimodal approaches that combine structured clinical information with unstructured medical image data have gained importance in osteoporosis research. However, existing work has so far focused primarily on binary classification tasks. For example, image features are extracted from fine-tuned CNNs on chest radiographs and fused with clinical variables that are transferred to the feature space via a multilayer perceptron (MLP). The combined features are then passed through another MLP to opportunistically screen for osteoporosis (Tang et al., 2025). A similar approach additionally uses dimensionality reduction via principal component analysis (PCA) and clustering-based feature selection to improve classification performance (Chagahi et al., 2024). For fracture risk classification, Schmarje et al. (2022) concatenated image features from pelvic radiographs with CRFs and used an MLP for binary prediction, without modeling the temporal dimension of fracture risk. Similarly, a small vision transformer (ViT-S) was pre-trained in a self-supervised manner and combined with clinical features processed via an MLP (Senanayake et al., 2023). Radiomics-based approaches extract features from CT images and fuse them with clinical variables, achieving improved predictive performance, particularly with gradient boosting models (Saravi et al., 2024; Zhang et al., 2024a). Only a few studies exist in the field of multimodal SA for osteoporotic fracture prediction. Shaikh et al. (2024) proposed a two-stage pipeline in which CNN-derived vertebral CT features are first used to classify 10-year fracture risk, and the resulting scores are then combined with age and BMI in a CoxPH model in the second stage. This approach achieved a C-index of 0.78 and demonstrated the potential of CNN-based fracture assessment, but is constrained by its two-stage design and the omission of certain CRFs. The most relevant study compared to our work was proposed by Kong et al. (2022), who introduced a two-stage multimodal survival modeling framework. Their approach combines ResNet-based spinal radiograph features with CRFs within a DeepSurv architecture. In the first stage, a keypoint detection model is trained to localize the vertebral bodies L1–L5 in spinal radiographs, which are then used to extract fixed regions of interest (ROIs) or corresponding image patches. These ROIs are kept fixed and serve as input to the second stage of the pipeline. In the second stage, the cropped ROIs are processed by a ResNet to extract image features, which are concatenated with CRFs and passed through a MLP, with DeepSurv acting as the survival prediction head. This approach demonstrated that multimodal methods incorporating temporal survival modeling can outperform traditional FRAX® and CoxPH baselines. The study of Kong et al. (2022) is limited by a small sample size, low event rate, single-center data, and potential selection bias and missing values. Moreover, due to reliance on simple feature concatenation, multimodal fusion is restricted to global-level integration, preventing explicit modeling of fine-grained cross-modal interactions. Motivated by these gaps in prior work, we propose the following approach to multimodal SA for osteoporotic fracture risk

assessment, leveraging pelvic radiographs as a more widely available alternative to DXA alongside recommended CRFs.

## 3. Methodology

In this section, we describe the proposed ATSF model, provide details about the used data, the model architecture along with our training procedure, and the evaluation setup. The code and model weights will be released upon acceptance.

### 3.1. Data

This work uses data from the Study of Osteoporotic Fractures (SOF), a population-based multicenter cohort study funded by the National Institutes of Health. Between 1986 and 2008, the study enrolled 9,704 predominantly Caucasian women aged 65 years or older across four U.S. clinical sites (Baltimore, Minneapolis, Portland, and Pittsburgh) and recorded longitudinal health information. The dataset includes radiographs from anatomically relevant regions for osteoporosis assessment, including pelvic images, along with CRFs, measurements obtained from blood samples, functional and cognitive tests, and precomputed BMD. It should be noted that DXA images used to derive BMD measurements are not used in the present study. Fracture status was systematically updated at four-month intervals throughout the study period (Cummings et al., 1995, 1998). The dataset is publicly accessible via the SOF online portal[1].

### 3.2. Baselines

To validate our model we compare it to three established baselines: First, we employ the WHO-endorsed FRAX® score, using the nine CRFs sex, age, BMI, smoking status, alcohol consumption, previous fractures, rheumatoid arthritis, glucocorticoid exposure, and parental history of hip fracture (Kanis et al., 2007). FRAX® is evaluated both with and without BMD, and we use the US (caucasian)-calibrated model to ensure consistency with the population of SOF. Second, we use a CoxPH model incorporating the same CRFs as FRAX® to ensure optimal comparability (Cox, 1972). Because the SOF dataset contains exclusively female participants, sex is excluded from this baseline due to its lack of discriminative value. The third baseline serves as our DL-baseline and follows the multimodal approach described by Kong et al. (2022). Since our work focuses on multimodal survival modeling rather than vertebral localization or ROI extraction, we base our DL-baseline exclusively on the second stage of the pipeline proposed by Kong et al. (2022). As the original publication does not specify the exact model variant, used weights, or layer parameters, our implementation adopts reasonable and commonly used choices to replicate the method as faithfully as possible. Specifically, we concatenate the eight normalized CRFs with 2,048 pelvic radiograph features extracted from a pretrained and frozen ResNet-50 with IMAGENET1K_V1 weights (He et al., 2016). The concatenated feature vector is fed to a trainable two-layer MLP ($2048+8 \rightarrow 1024 \rightarrow \text{ReLU} \rightarrow 512$) and subsequently passed into a DCTM prediction head. While the original study employed DeepSurv (Katzman et al., 2018), we deliberately replace this component with a DCTM head (12 Bernstein

---

1. https://sofonline.ucsf.edu/

polynomials) to ensure comparability across survival models and to isolate differences in feature extraction and multimodal fusion rather than survival-head design (Campanella et al., 2025). The MLP used for feature fusion remains identical to the original architecture. The DL-baseline model follows the same optimization strategy, training protocol, and final hyperparameter settings as specified in Section 3.4; the only difference is that these hyperparameters are manually selected rather than determined via automatic hyperparameter optimization.

## 3.3. Attention-to-Survival Fusion Model

Our ATSF model for multimodal SA of hip-fracture risk consists of four components (Figure 1): the image encoder (green), the CRF encoder (red), the fusion layers (blue), and the DCTM head (orange). In the image encoder, pelvic radiographs are used intentionally because, unlike DXA scans, they are widely available and thus enable broader applicability (Hsieh et al., 2021; Kanis et al., 2021; Kanis and Johnell, 2005). Depending on the configuration, the model operates on unilateral (left hip), bilateral (left plus mirrored right hip), or full pelvis radiographs acquired at the first visit. The latter is incorporated via a MedVAE-derived projection that compresses the image while preserving the most relevant structural information (Varma et al., 2025). Features are then extracted using a Vision Transformer (ViT, pre-trained on ImageNet, vit_base_patch16_224 without classification head) and reduced to the most informative dimensions via a Q-Former, ensuring that only the features most relevant for risk prediction are retained (Wu et al., 2020; Deng et al., 2009; Zhang et al., 2024b). In the CRF encoder, the eight CRFs used in the CoxPH and DL baselines are embedded via learnable embedding layers. This ensures that structured clinical information is represented in a comparable latent space alongside high-dimensional image features. The fusion layers serve as the central mechanism for integrating the two modalities. A randomly initialized, task-specific programmable token (query) iteratively aggregates relevant information from the multimodal features (keys and values) through three successive attention mechanisms, forming an explicit latent representation of the target risk. Stacking three fusion layers allows the model to refine this representation and capture complex relationships between modalities. Finally, the representative token is passed to the DCTM head, which performs SA of hip-fracture risk. The general shift model with 6–14 Bernstein polynomials is used to approximate the logistic function (Campanella et al., 2025).

## 3.4. Training Details

Prior to training, a keypoint-based detection algorithm (Damm et al., 2022) was applied to standardize the pelvic radiograph orientation, isolate the left and right hip regions, and crop images to the model input size of $224 \times 224$ pixels. The algorithm achieves an accuracy of $94\%$ for the left hip and $96\%$ for the right hip. Images for which keypoint detection failed were excluded. Full-pelvis radiographs were additionally processed using MedVAE pretrained on radiographs to generate a single-channel projection with an eight-fold compression per dimension (Varma et al., 2025). Because the hip region is nearly square, the resulting projection was rescaled to the required input size, accepting minor geometric distortion to avoid loss of relevant anatomical information. Normalization was applied to all

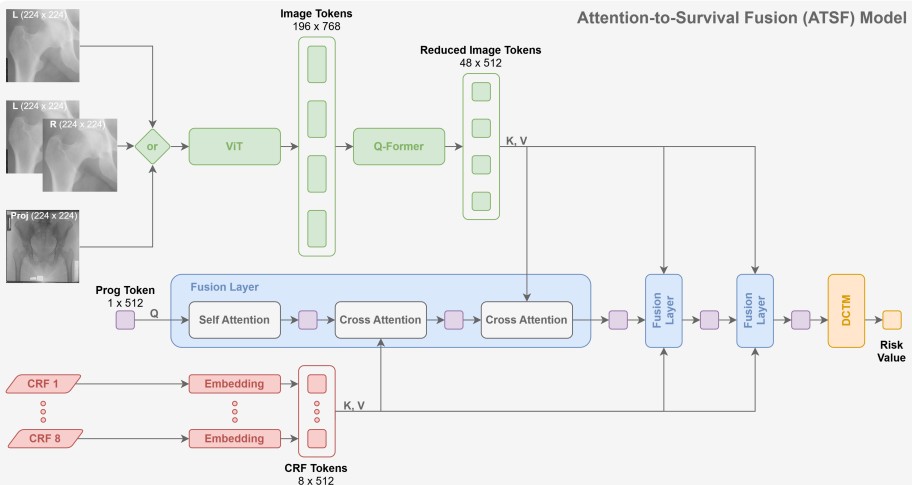

Figure 1: Overview of the ATSF model for hip-fracture risk prediction. Radiographic features (green) and CRFs (red) are integrated through multiple fusion layers (blue), enabling modality-aware weighting. The combined representation is passed to a DCTM head (orange) for time-dependent risk estimation.

preprocessed input images. Given that the SOF dataset originates from multiple clinical sites, we also assessed center-specific differences in imaging protocols or acquisition quality. While handling such heterogeneity was not the primary focus of this study, we observed comparable distributions across centers without pronounced biases. Based on this assessment, no additional harmonization or center-specific normalization methods were applied. Continuous CRFs were standardized, and categorical CRFs were one-hot encoded for the ATSF model, with variable names being encoded for clear identification. For the DL-baseline, categorical variables were included as binary indicators (0 or 1) and continuous variables were standardized. We evaluated several alternative imputation strategies for missing CRF values (mean, median, and embedding-based imputation (see Appendix E Tables 8, 9, 10 and 11)). As none of these strategies resulted in statistically significant performance differences compared to the original setup, missing CRF values in the DL-baseline were encoded as -1. For ATSF, we considered different strategies for handling missing values, including a learnable representation of missingness within the attention mechanism and a masking-based approach in which missing entries are excluded from learning. The masking-based strategy was adopted, as it is consistent with the assumption of non-informative missingness. After preprocessing, 7,818 of the 9,704 SOF participants had both a valid radiograph and complete time-to-event information. These participants were randomly assigned to training, validation, and test sets in a 50/10/40 ratio, ensuring comparable distributions of key variables and sufficient fracture events within each split (e.g., 14.2 % in the test set, see Appendix B). Smaller test sets would have led to unstable and high-variance metric estimates, potentially resulting in misleading conclusions regarding model performance. We therefore prioritized a robust assessment on a large held-out test set. The split was performed externally by an independent collaborator, and was held out to prevent any optimization on the test data. Since the externally created test set contains missing values in several CRFs and

classic methods such as FRAX® and CoxPH cannot process these, we test on a reduced, fully observed subset, referred to as Test$_{\text{reduced}}$. This ensures a fair comparison of all methods using exactly the same samples. The complete test set (referred to as Test$_{\text{complete}}$), including missing values, was later used in the ablation experiments (see Section 3.5). To improve robustness, the training set was augmented using random Gaussian noise, Gaussian smoothing, affine transformations, random contrast adjustment, and zoom operations. All models were optimized using AdamW with a weight decay of $10^{-2}$ and a batch size of 169. The majority of experiments were performed on the NVIDIA H100 NVL (96 GB VRAM) while training with lower computational requirements was also executed on GPUs with lower memory capacity such as the NVIDIA Tesla V100 SXM2 or NVIDIA RTX 6000 Ada Generation. The DL-baseline was trained for 300 epochs with a learning rate of $7.2 \times 10^{-5}$. For the ATSF model, we performed hyperparameter optimization (HPO) with 150 epochs over learning rates between $1 \times 10^{-8}$ and $5 \times 10^{-6}$, augmentation probabilities, the number of Bernstein polynomials (6–14), the number of fusion layers, attention heads (8 or 16), and the number of Q-Former tokens (16–64). All DL-based survival models were trained using the NLL loss, and model selection was conducted based on the validation C-index.

### 3.5. Evaluation and Ablation

To quantify the predictive accuracy of our models in comparison to the baselines, we primarily use the C-index. Introduced in 1982 by Harrell et al., this metric is based on rank correlations between predicted and observed outcomes and is formally defined as the ratio of all evaluable patient pairs for which the prediction and the actual observed outcome are concordant (Harrell et al., 1982, 1996). In addition to the C-index, we use the AUC to quantify the discriminatory power of the models independently of the ranking. Since, unlike the C-index, the AUC does not take into account the temporal aspect of the event occurrence, the test data set is adjusted for this analysis. Participants who suffer from a fracture after a predefined period of time are coded as negatives, and right-censored observations are excluded from the AUC calculation (Hanley and McNeil, 1982). Due to these limitations and the required adjustments, the AUC should be interpreted with caution and is considered a secondary metric complementing the C-index. Following the recommendation of Harrell et al., we perform bootstrapping for model comparison by resampling the test data with replacement (2000 bootstrap samples) (Harrell et al., 1996). Specifically, we use the nonparametric test by Kang et al. (2015), which allows robust estimates of the uncertainty of performance measures, including the calculation of 95 % confidence intervals. All statistical analyses between the best ATSF and baseline model are performed on the independent hold-out test set unless explicitly stated otherwise. A two-sided significance level of $p < 0.05$ is used as a reference threshold. Results are reported for 5- and 10-year time horizons. The 10-year horizon matches the FRAX® assessment period, while the shorter 5-year horizon is close to clinical guideline recommendations (3 years according to DVO (2026)) and ensures sufficient positive cases for stable evaluation.

To interpret the ATSF model, we analyze the cross-attention weights of the programmable token within the fusion blocks, following the general approach used in attention-based survival models such as SurvTrace (Wang and Sun, 2022). For the CRFs, we compute cross-attention between the programmable token (query) and all CRF tokens (keys). For each

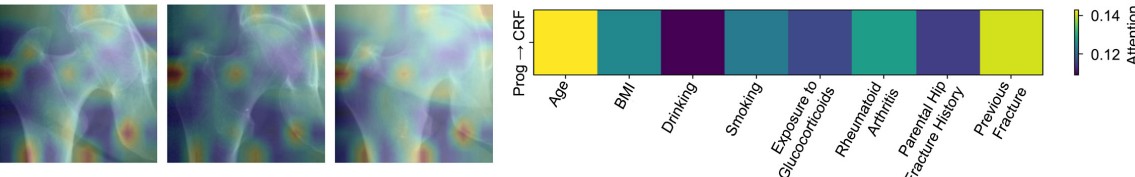

Figure 2: Visual (left) and CRF attention (right) of the ATSF model.

attention head, this yields a distribution indicating how strongly the programmable token attends to each CRF token. We then average the attention weights across heads and extract the row corresponding to the programmable token, resulting in a one-dimensional vector with one value per CRF token. Visual attention maps are obtained analogously. After processing the images with the ATSF image encoder, the resulting visual tokens form a 2D spatial grid. We compute cross-attention from the programmable token to all visual tokens, average the weights across heads, and extract the programmable-token query row. The resulting weights are reshaped to the original 2D token grid, upsampled to image resolution using bilinear interpolation, and normalized for visualization.

To assess the contribution of each modality and image configuration, we conducted a structured ablation study. First, we evaluated unimodal variants of the proposed architecture using only CRFs or only radiograph features, implemented by disabling the complementary modality while keeping all other model components identical. Second, we examined the influence of different radiographic inputs by training separate models using unilateral (left) radiographs, bilateral (left and mirrored right) radiographs, and full projection sets. These experiments allow us to isolate the effect of each data source and to quantify the added value of multimodal integration compared to unimodal or reduced-input variants.

Because FRAX® and CoxPH cannot process missing values, whereas the ATSF model explicitly supports them, we additionally evaluated ATSF under two test set conditions: a reduced test set from which all missing values were removed ($Test_{reduced}$) and the full test set containing missing values ($Test_{complete}$). This design allows us to disentangle several effects: the potential advantage of a larger evaluable test set enabled by missing-value handling, the ATSF model's performance when evaluated on exactly the same data as FRAX and CoxPH, and the sensitivity of the ATSF model to missing information.

## 4. Results

The C-index and AUC results of the different models on $Test_{reduced}$ with a 10-year time horizon are summarized in Table 1. The table also reports 95 % confidence intervals and statistical significance against CoxPH with $CRF_{BMD}$, which is marked by *.

Based on both C-index and AUC, the inclusion of BMD is consistently associated with improved performance for FRAX® and CoxPH. Among these clinical baselines, CoxPH with $CRF_{BMD}$ achieves the highest discrimination, with a C-index of 0.769 and an AUC of 0.788, outperforming all baseline models tested. Within the DL-Baseline, the best-performing configurations are observed for bilateral pelvic radiographs followed by projection-based inputs. Across all multimodal settings, the ATSF model yields higher performance than the corresponding configurations of the DL-baseline. The bilateral pelvic radiograph con-

Table 1: Performance comparison of all models on $\text{Test}_{\text{reduced}}$ using C-index, AUC, and their respective 95 % confidence intervals (10-year time horizon). $\text{CRF}_{\text{BMD}}$ denotes inclusion of BMD. For multimodal (MM) and image-only models, subscripts indicate the radiograph configuration: L for unilateral (left hip), LR for bilateral (left & mirrored right hip), and Proj for the MedVAE-derived full-pelvis projection.

| Model | Modality | $\uparrow$ **C-Index** | $\uparrow$ **AUC** |
|---|---|---|---|
| FRAX® | CRF | 0.703 (0.652 - 0.753) | 0.718 (0.667 - 0.768) |
| | $\text{CRF}_{\text{BMD}}$ | 0.746 (0.702 - 0.787) | 0.758 (0.716 - 0.799) |
| CoxPH | CRF | 0.724 (0.674 - 0.772) | 0.746 (0.696 - 0.794) |
| | $\text{CRF}_{\text{BMD}}$ | 0.769 (0.727 - 0.808) | 0.788 (0.746 - 0.828) |
| DL-Baseline | $\text{MM}_{\text{L}}$ | 0.716 (0.662 - 0.765) | 0.737 (0.684 - 0.758) |
| | $\text{MM}_{\text{LR}}$ | 0.752 (0.706 - 0.796) | 0.776 (0.731 - 0.818) |
| | $\text{MM}_{\text{Proj}}$ | 0.731 (0.685 - 0.775) | 0.755 (0.710 - 0.799) |
| ATSF (Ours) | CRF | 0.722 (0.673 - 0.771) | 0.742 (0.690 - 0.792) |
| | $\text{Image}_{\text{L}}$ | 0.758 (0.711 - 0.803) | 0.775 (0.726 - 0.820) |
| | $\text{Image}_{\text{LR}}$ | 0.757 (0.715 - 0.795) | 0.771 (0.728 - 0.812) |
| | $\text{Image}_{\text{Proj}}$ | 0.709 (0.660 - 0.757) | 0.726 (0.676 - 0.774) |
| | $\text{MM}_{\text{L}}$ | 0.770 (0.736 - 0.836) | 0.788 (0.720 - 0.815) |
| | $\text{MM}_{\text{LR}}$ | **0.805\*** (0.764 - 0.844) | **0.826** (0.786 - 0.863) |
| | $\text{MM}_{\text{Proj}}$ | 0.742 (0.699 - 0.782) | 0.762 (0.715 - 0.803) |

figuration achieves the highest overall discrimination, reaching a C-index of 0.805 and an AUC of 0.826. It demonstrates statistically significant improvement compared to CoxPH with $\text{CRF}_{\text{BMD}}$. For image-only variants, ATSF shows higher performance than the baseline models, with the exception of CoxPH with $\text{CRF}_{\text{BMD}}$. In contrast, the text-only variant of ATSF attains a C-index of 0.722 and exhibits lower discrimination than CoxPH when evaluated with the same input variables. Analysis of the input modalities shows that bilateral pelvic radiographs always achieve better C-indices and AUC than projections. Unilateral is only preferable to projections in certain configurations. Consistent patterns are observed for the 5-year prediction horizon on $\text{Test}_{\text{reduced}}$, where the multimodal ATSF model with bilateral inputs again achieves superior evaluation metrics compared to alternative approaches (see Appendix E Table 5). Unlike with a 10-year time horizon, it can be seen here that ATSF outperforms the corresponding CoxPH model with CRFs only. Additional Kaplan-Meier analysis of the best baseline (CoxPH $\text{CRF}_{\text{BMD}}$) and overall best ATSF configuration ($\text{MM}_{\text{LR}}$) highlights that both models achieve a comparably strong separation between low- and high-risk participants (see Appendix E, Figures 4 and 5). The results on $\text{Test}_{\text{complete}}$ (see Appendix E, Tables 6 and 7) show no substantial changes in overall performance compared with $\text{Test}_{\text{reduced}}$, even with standard techniques for missing values. Across both test sets, the ATSF model exhibits higher values for the evaluated metrics when using a 5-year prediction horizon. To contextualize the ATSF model outputs, we further examine its attention mechanisms, illustrated in Figure 2. The overlaid example images of fracture cases indicate that the model consistently allocates higher attention to regions including

the greater trochanter, the femoral neck (Ward's triangle), and adjacent areas of the lower hip. The cross-attention between CRFs and the programmable token shows the largest contributions from age and prior fracture history, while alcohol consumption receives the lowest attention values. The hyperparameters of all models are reported in Appendix C.

## 5. Discussion

Based on the results of the two established baselines FRAX® and CoxPH, it can initially be deduced that the information gained by adding the DXA-determined BMD provides predictive added value. CoxPH outperforms FRAX® with regard to SA based on the SOF data set. Apart from the CoxPH model with $CRF_{BMD}$, the multimodal bilateral DL-baseline, which was trained on pelvic radiographs and CRFs excluding BMD, is better than all other baselines. However, since pelvic radiographs are more widely available than DXA, this baseline is still preferable in some scenarios. The ATSF model presented in this paper outperforms all other approaches in its bilateral multimodal configuration. It can therefore be concluded that the inclusion of right radiographs offers an advantage in this application. In comparison, the general lower performance of the MedVAE full-pelvis projections suggests that not all relevant trabecular structures are preserved by MedVAE and that the slight geometric distortion negatively affects predictive accuracy. Ablation studies further prove that only the combination of CRFs and images represents a significant improvement over established methods. Although BMD is also extracted from an image modality, the ATSF model in its best configuration is superior to established methods with $CRF_{BMD}$. This suggests that the model is capable of delivering higher predictive performance from the previously unrecommended combination of CRFs and pelvic radiographs. Since the results on the full test set $Test_{complete}$ do not show notable degradation compared with $Test_{reduced}$, the findings indicate that the model handles missing values in the CRFs robustly and maintains stable predictive performance even under incomplete clinical information. Slight performance differences, e.g., at the 5-year horizon, may be related to missing values in influential CRFs (e.g., age), which could reduce prediction accuracy for affected participants. While this effect cannot be precisely quantified, it is consistent with our feature importance analysis. The observed improvement in performance at shorter prediction horizons aligns with the notion that radiographic features capture structural risk characteristics whose predictive relevance may diminish over longer time spans due to ongoing bone remodeling. In addition, several CRFs are time-varying, and their influence is inherently more stable over shorter horizons, which may further contribute to the superior performance observed at 5 years compared with 10 years. The attention patterns provide additional context for interpreting the model's behavior. The model consistently places emphasis on anatomically relevant regions of the proximal femur and prioritizes key CRFs, such as age and prior fractures. These patterns suggest that the model leverages information consistent with established risk factors for osteoporotic fractures, supporting plausibility and interpretability.

## 6. Conclusion, Limitations, and Future Work

In this work, we introduced the ATSF model, a multimodal survival analysis framework that integrates pelvic radiographs with CRFs to predict long-term hip-fracture risk. Across

all evaluated configurations, ATSF consistently outperformed established baselines, with the bilateral multimodal variant achieving the highest performance on both reduced and complete test sets. These results demonstrate that pelvic radiographs – an imaging modality far more widely available than DXA – provide clinically valuable structural information that, when combined with CRFs in a unified survival framework, can lead to substantially improved fracture risk estimation. In addition to its performance improvements, ATSF provides modality-specific interpretability, which allows clinicians to understand the relative contribution of radiographic and clinical features. These results position ATSF as a promising step toward more accessible, accurate, and individualized osteoporosis risk assessment, leveraging pelvic radiographs as a widely available alternative to the current standard of DXA imaging.

Despite promising results, this study has limitations. The SOF cohort includes only US women, with few fracture events, and potential selection bias cannot be excluded. The evaluation metrics used have known limitations, especially under high censoring. No k-fold cross-validation was performed, and treatment history, a relevant confounder, was not available. The DCTM survival head was used as a fixed reference and not systematically compared with alternative architectures. Furthermore, while an extensive ablation study was performed, it does not isolate the contributions of every individual model component. ATSF variants including DXA imaging or BMD were also not evaluated. Visual attention maps are only an approximate visualization, as the underlying queries are freely learnable rather than pixel-bound, making it suitable for intuition but not precise localization. Moreover, the DL-baseline hyperparameters were selected manually based on validation performance and training stability, rather than through systematic automated hyperparameter optimization.

Potential avenues for future work are to address these limitations by improving generalizability through integration of additional cohorts such as MrOS (Orwoll et al., 2005; Blank et al., 2005), incorporating alternative or complementary evaluation metrics for more differentiated analysis, considering additional factors such as DXA-derived BMD and treatment history, and performing more detailed ablation studies to disentangle the contributions of individual model components.

## Acknowledgments

This project was supported by the modular AI Imaging Pipelines (mAIPipes) Grant, Application No. 22024025 KI-Förderrichtlinie Schleswig-Holstein, Germany. Furthermore, this work was supported by the IDIR-Project (Digital Implant Research), a cooperation financed by Kiel University, University Hospital Schleswig-Holstein and Helmholtz Zentrum Hereon.

The Study of Osteoporotic Fractures (SOF) is supported by National Institutes of Health funding. The National Institute on Aging (NIA) provides support under the following grant numbers: R01 AG005407, R01 AR35582, R01 AR35583, R01 AR35584, R01 AG005394, R01 AG027574, and R01 AG027576. We also would like to thank Ida Häggström for her support on questions related to DCTMs.

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

## Appendix A. Theoretical Foundations of Survival Analysis

The central theoretical concepts of survival analysis (SA) are the survival function $S_T(t)$ and the hazard function $h_T(t)$. The former describes the probability that an event has not occurred by a certain point $t$ in time $T$ (assumed to be continuous here) and is formally defined as

$$S_T(t) := P(T > t) = 1 - F_T(t). \tag{1}$$

In this case, $F_T(t)$ describes the cumulative distribution function defined as $F_T(t) := P(T \leq t)$. The hazard rate (or hazard function)

$$h_T(t) := \lim_{\Delta t \to 0} \frac{1}{\Delta t} P(t \leq T < t + \Delta t | T \geq t) = \frac{f_T(t)}{S_T(t)} \tag{2}$$

describes the instantaneous risk of the event occurring at time $t$, provided that it has not yet occurred by that time. It thus links the survival time density function $f_T(t)$ with the survival function $S_T(t)$ and provides a continuous, time-dependent representation of the risk. The cumulative hazard function $H_T(t)$ is often used as an intermediate measure, which describes the total risk up to time $t$ and is calculated using

$$H_T(t) := \int_0^t h_T(u) du = -\log(S_T(t)). \tag{3}$$

This relationship illustrates the close functional connection between hazard and survival functions and forms a central basis for many parametric, semiparametric, and deep learning (DL)-based methods of modern SA (Collett, 2023; Wiegrebe et al., 2024; George et al., 2014).

Data characteristics are crucial for selecting and developing suitable models in SA. The time dependence of covariates is particularly relevant: variables whose values change over time (e.g., lifestyle factors) are referred to as time-varying features (TVFs), while time-varying effects (TVEs) describe the time-varying influences of characteristics on risk or hazard rates. Both require models that go beyond the assumption of proportional hazards. Similarly, the dimensionality and modality of the data influence the choice of model. High-dimensional and multimodal data sets, such as those from clinical, molecular, and imaging sources, require specialized methods that can combine structured and unstructured information. Finally, the types of outcomes must also be taken into account: censoring (incompletely observed events), truncation (subjects are not part of the data set), competing risks, multi-state scenarios, and recurrent events pose different statistical challenges and require flexible modeling approaches (Wiegrebe et al., 2024).

To model the distribution of event times based on the entry time, exit time, event indicator, and a feature vector, various methodological approaches are used in SA (Collett, 2023; Wiegrebe et al., 2024; George et al., 2014). These can be broadly categorized into three main groups:

- **Non-parametric methods:** Do not assume any specific distribution of survival times (Collett, 2023). Typical examples include the Kaplan–Meier estimator (Kaplan and Meier, 1958) and the log-rank test (Savage, 1956).

- **Semi-parametric methods:** Model the effect of covariates without assuming a specific baseline hazard. The Cox proportional hazards model (CoxPH) estimates covariate effects via hazard ratios under the assumption of proportional hazards and uses the partial likelihood for parameter estimation (Cox, 1972; George et al., 2014; Wiegrebe et al., 2024).

- **Parametric methods:** Assume a specific hazard or survival distribution (e.g., exponential or Weibull) and include proportional and additive hazard models, accelerated failure time (AFT) models, and piecewise exponential models (PEMs) (Collett, 2023; George et al., 2014; Wiegrebe et al., 2024).

With the rapid advances in machine learning (ML) and DL, SA has significantly benefited from the increasing modeling capacity of these approaches. The integration of multi-modal data sources and the growing emphasis on model interpretability have further contributed to the rising popularity of DL-based SA methods. Early neural network–based extensions of classical statistical approaches emerged in the mid-1990s and continue to build upon traditional SA frameworks while leveraging the representational flexibility of modern architectures. Feedforward neural networks (FFNNs), for instance, enable flexible estimation of (semi-)parametric hazard functions and can account for TVFs. Convolutional neural networks (CNNs), often applied through transfer learning, facilitate the inclusion of unstructured data such as medical images, whereas recurrent neural networks (RNNs) are particularly suited for modeling temporal dependencies and longitudinal data. Autoencoder-based architectures are commonly used for dimensionality reduction and feature representation. Wiegrebe et al. (Wiegrebe et al., 2024) provide a comprehensive review of recent DL-based survival methods, systematically categorizing them according to model class, loss function, and parameterization, as well as their supported survival tasks and interpretability characteristics. Selected approaches are discussed in more detail in Section 2.

## Appendix B. Study of Osteoporotic Fractures (SOF): Cohort Characteristics

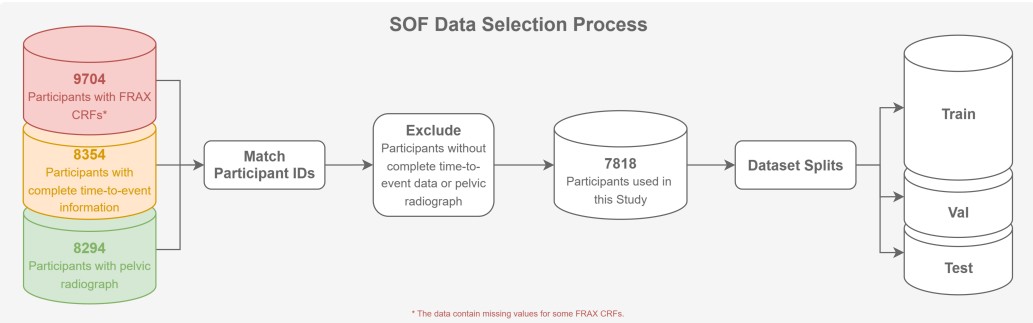

Figure 3: Overview of the SOF data selection process used in this study.

Table 2: Dataset characteristics of the training, validation and test sets derived from SOF. Due to missing values for some variables, percentages do not always total 100 %.

| Characteristics | Overall | | Train | | Validation | | Test_complete | | Test_reduced | |
|---|---|---|---|---|---|---|---|---|---|---|
| n participants | 7,818 (100.0%) | | 3,740 (47.8%) | | 931 (11.9%) | | 3,147 (40.3%) | | 1,969 (25.2%) | |
| Age (y) | 71.7 ± 5.2 [65, 90] | | 71.7 ± 5.3 [65.0, 90.0] | | 71.7 ± 5.1 [65.0, 90.0] | | 71.6 ± 5.1 [65, 90] | | 71.4 ± 5.0 [65, 90] | |
| BMI | 26.5 ± 4.7 [14.3, 58.4] | | 26.4 ± 4.6 [14.3, 49.7] | | 26.6 ± 4.5 [16.0, 46.0] | | 26.6 ± 4.8 [15.2, 58.4] | | 26.6 ± 4.7 [15.2, 58.4] | |
| | **Yes** | **No** | **Yes** | **No** | **Yes** | **No** | **Yes** | **No** | **Yes** | **No** |
| Smoking | 782 (10.0%) | 7,011 (89.7%) | 365 (9.8%) | 3,366 (90.0%) | 93 (10.0%) | 833 (89.5%) | 324 (10.3%) | 2,812 (89.4%) | 195 (9.9%) | 1,774 (90.1%) |
| Drinking | 248 (3.2%) | 7,565 (96.8%) | 118 (3.2%) | 3,620 (96.8%) | 36 (3.9%) | 895 (96.1%) | 94 (3.0%) | 3,050 (96.9%) | 50 (2.5%) | 1,919 (97.5%) |
| Previous Fracture | 2,844 (36.4%) | 4,928 (63.0%) | 1,376 (36.8%) | 2,338 (62.5%) | 339 (36.4%) | 587 (63.1%) | 1,129 (35.9%) | 2,003 (63.6%) | 663 (33.7%) | 1,306 (66.3%) |
| Rheumatoid Arthritis | 505 (6.5%) | 7,176 (91.8%) | 229 (6.1%) | 3,447 (92.2%) | 64 (6.9%) | 849 (91.2%) | 212 (6.7%) | 2,880 (91.5%) | 129 (6.6%) | 1,840 (93.4%) |
| Glucocorticoid Exposure | 896 (11.5%) | 6,768 (86.6%) | 436 (11.7%) | 3,233 (86.4%) | 115 (12.4%) | 797 (85.6%) | 345 (11.0%) | 2,738 (87.0%) | 227 (11.5%) | 1,742 (88.5%) |
| Parental Hip Fracture | 879 (11.2%) | 5,485 (70.2%) | 427 (11.4%) | 2,641 (70.6%) | 113 (12.1%) | 625 (67.1%) | 339 (10.8%) | 2,219 (70.5%) | 266 (13.5%) | 1,703 (86.5%) |
| Hip Fracture | 1,131 (14.5%) | 6,687 (85.5%) | 550 (14.7%) | 3,190 (85.3%) | 135 (14.5%) | 796 (85.5%) | 446 (14.2%) | 2,701 (85.8%) | 285 (14.5%) | 1,684 (85.5%) |
| Time to Fracture (d) | 5,120.3 ± 2,275.2 [6, 8,535] | | 5,132.2 ± 2,277.1 [6, 8,521] | | 5,081.0 ± 2,286.4 [6, 8,520] | | 5,117.9 ± 2,269.6 [20, 8,535] | | 5,418.2 ± 2,108.6 [20, 8,535] | |
| Hip Fracture <10y | 499 (6.4%) | 7,319 (93.6%) | 238 (6.4%) | 3,502 (93.6%) | 62 (6.7%) | 869 (93.3%) | 199 (6.3%) | 2,948 (93.7%) | 115 (5.8%) | 1,854 (94.2%) |
| Time to Fracture <10y (d) | 2,155.1 ± 980.6 [6, 3,650] | | 2,165.0 ± 972.4 [6, 3,650] | | 2,132.7 ± 1,020.3 [6, 3,648] | | 2,150.2 ± 977.9 [20, 3,642] | | 2,378.6 ± 818.9 [20, 3,642] | |

## Appendix C. ATSF Hyperparameters

Table 3: Overview of the optimal hyperparameters obtained from the HPO across all ATSF model configurations.

| Hyperparameter | CRF | Image | | | MM | | |
|---|---|---|---|---|---|---|---|
| | | L | LR | Proj | L | LR | Proj |
| Learning Rate $r$ | 2.4e-7 | 2.0e-7 | 6.4e-7 | 1.1e-6 | 1.7e-7 | 7.2e-7 | 1.8e-6 |
| $p$ Affine | - | 0.5 | 0.5 | 0.0 | 0.8 | 0.5 | 0.8 |
| $p$ Contrast | - | 0.0 | 0.0 | 0.5 | 0.0 | 0.0 | 0.5 |
| $p$ Gaussian Noise | - | 0.0 | 0.0 | 0.9 | 0.9 | 0.5 | 0.5 |
| $p$ Gaussian Smooth | - | 0.3 | 0.3 | 0.5 | 0.0 | 0.3 | 0.0 |
| $p$ Zoom | - | 0.8 | 0.5 | 0.5 | 0.5 | 0.5 | 0.5 |
| $n$ Bernstein Polynomials | 10 | 12 | 10 | 12 | 10 | 14 | 12 |
| $n$ Fusion Layers | - | - | - | - | 1 | 3 | 6 |
| $n$ Attention Heads | - | - | - | - | 16 | 8 | 16 |
| $n$ Q-Former Tokens | - | - | - | - | 32 | 64 | 32 |

## Appendix D. Baseline Hyperparameters

Table 4: Overview of the hyperparameters used for the DL-baseline models.

| Hyperparameter | Value |
|---|---|
| Learning Rate $r$ | 7.2e-5 |
| $p$ Affine | 0.8 |
| $p$ Contrast | 0.0 |
| $p$ Gaussian Noise | 0.5 |
| $p$ Gaussian Smooth | 0.5 |
| $p$ Zoom | 0.5 |
| $n$ Bernstein Polynomials | 12 |

# Appendix E. Additional Results

Table 5: Performance comparison of all models on Test$_{\text{reduced}}$ using C-index, AUC, and their respective 95 % confidence intervals (5-year time horizon). CRF$_{\text{BMD}}$ denotes inclusion of BMD. For multimodal (MM) and image-only models, subscripts indicate the radiograph configuration: L for unilateral (left hip), LR for bilateral (left & mirrored right hip), and Proj for the MedVAE-derived full-pelvis projection.

| Model | Modality | ↑ C-Index | ↑ AUC |
|---|---|---|---|
| FRAX® | CRF | 0.675 (0.556 - 0.771) | 0.680 (0.576 - 0.772) |
| | CRF$_{\text{BMD}}$ | 0.751 (0.663 - 0.829) | 0.755 (0.670 - 0.831) |
| CoxPH | CRF | 0.681 (0.579 - 0.779) | 0.687 (0.586 - 0.779) |
| | CRF$_{\text{BMD}}$ | 0.764 (0.667 - 0.840) | 0.769 (0.685 - 0.845) |
| DL-Baseline | MM$_{\text{L}}$ | 0.705 (0.604 - 0.796) | 0.707 (0.604 - 0.796) |
| | MM$_{\text{LR}}$ | 0.721 (0.636 - 0.798) | 0.727 (0.641 - 0.802) |
| | MM$_{\text{Proj}}$ | 0.706 (0.627 - 0.785) | 0.714 (0.629 - 0.789) |
| ATSF (Ours) | CRF | 0.696 (0.600 - 0.784) | 0.702 (0.608 - 0.790) |
| | Image$_{\text{L}}$ | 0.775 (0.689 - 0.855) | 0.778 (0.684 - 0.863) |
| | Image$_{\text{LR}}$ | 0.758 (0.679 - 0.827) | 0.761 (0.682 - 0.831) |
| | Image$_{\text{Proj}}$ | 0.748 (0.679 - 0.813) | 0.753 (0.682 - 0.817) |
| | MM$_{\text{L}}$ | 0.782 (0.701 - 0.853) | 0.785 (0.699 - 0.860) |
| | MM$_{\text{LR}}$ | **0.808*** (0.726 - 0.881) | **0.813** (0.732 - 0.882) |
| | MM$_{\text{Proj}}$ | 0.758 (0.683 - 0.827) | 0.762 (0.691 - 0.829) |

Table 6: Performance comparison of all models on Test$_{\text{complete}}$ using C-index, AUC, and their respective $95\%$ confidence intervals (10-year time horizon). CRF$_{\text{BMD}}$ denotes inclusion of BMD. For multimodal (MM) and image-only models, subscripts indicate the radiograph configuration: L for unilateral (left hip), LR for bilateral (left and mirrored right hip), and Proj for the MedVAE-derived full-pelvis projection. For FRAX® and CoxPH, missing categorical variables were set to 0. Missing continuous variables in the CoxPH model were imputed by the mean.

| Model | Modality | ↑ C-Index | ↑ AUC |
|---|---|---|---|
| FRAX® | CRF | 0.708 (0.666 - 0.749) | 0.724 (0.680 - 0.767) |
| | CRF$_{\text{BMD}}$ | 0.746 (0.708 - 0.784) | 0.760 (0.721 - 0.797) |
| CoxPH | CRF | 0.719 (0.675 - 0.760) | 0.740 (0.695 - 0.782) |
| | CRF$_{\text{BMD}}$ | 0.757 (0.718 - 0.796) | 0.777 (0.739 - 0.813) |
| DL-Baseline | MM$_{\text{L}}$ | 0.717 (0.673 - 0.759) | 0.740 (0.697 - 0.782) |
| | MM$_{\text{LR}}$ | 0.741 (0.702 - 0.780) | 0.765 (0.722 - 0.804) |
| | MM$_{\text{Proj}}$ | 0.731 (0.691 - 0.770) | 0.754 (0.713 - 0.793) |
| ATSF (Ours) | CRF | 0.721 (0.677 - 0.763) | 0.742 (0.698 - 0.784) |
| | Image$_{\text{L}}$ | 0.761 (0.721 - 0.800) | 0.779 (0.738 - 0.818) |
| | Image$_{\text{LR}}$ | 0.757 (0.715 - 0.795) | 0.771 (0.728 - 0.812) |
| | Image$_{\text{Proj}}$ | 0.708 (0.667 - 0.748) | 0.724 (0.680 - 0.765) |
| | MM$_{\text{L}}$ | 0.771 (0.731 - 0.811) | 0.789 (0.747 - 0.829) |
| | MM$_{\text{LR}}$ | **0.800\*** (0.764 - 0.834) | **0.821** (0.786 - 0.856) |
| | MM$_{\text{Proj}}$ | 0.736 (0.698 - 0.774) | 0.756 (0.715 - 0.796) |

Table 7: Performance comparison of all models on Test$_{\text{complete}}$ using C-index, AUC, and their respective 95 % confidence intervals (5-year time horizon). CRF$_{\text{BMD}}$ denotes inclusion of BMD. For multimodal (MM) and image-only models, subscripts indicate the radiograph configuration: L for unilateral (left hip), LR for bilateral (left and mirrored right hip), and Proj for the MedVAE-derived full-pelvis projection. For FRAX® and CoxPH, missing categorical variables were set to 0. Missing continuous variables in the CoxPH model were imputed by the mean.

| Model | Modality | ↑ C-Index | ↑ AUC |
|---|---|---|---|
| FRAX® | CRF | 0.683 (0.597 - 0.767) | 0.687 (0.599 - 0.771) |
| | CRF$_{\text{BMD}}$ | 0.741 (0.664 - 0.815) | 0.744 (0.668 - 0.815) |
| CoxPH | CRF | 0.692 (0.605 - 0.776) | 0.692 (0.603 - 0.769) |
| | CRF$_{\text{BMD}}$ | 0.744 (0.655 - 0.821) | 0.748 (0.667 - 0.824) |
| DL-Baseline | MM$_{\text{L}}$ | 0.702 (0.618 - 0.784) | 0.708 (0.625 - 0.787) |
| | MM$_{\text{LR}}$ | 0.729 (0.656 - 0.794) | 0.733 (0.659 - 0.802) |
| | MM$_{\text{Proj}}$ | 0.723 (0.654 - 0.789) | 0.728 (0.654 - 0.794) |
| ATSF (Ours) | CRF | 0.695 (0.614 - 0.778) | 0.699 (0.608 - 0.781) |
| | Image$_{\text{L}}$ | 0.772 (0.690 - 0.846) | 0.776 (0.691 - 0.853) |
| | Image$_{\text{LR}}$ | 0.758 (0.679 - 0.827) | 0.761 (0.682 - 0.831) |
| | Image$_{\text{Proj}}$ | 0.752 (0.689 - 0.813) | 0.756 (0.686 - 0.819) |
| | MM$_{\text{L}}$ | 0.783 (0.707 - 0.852) | 0.787 (0.707 - 0.852) |
| | MM$_{\text{LR}}$ | **0.817\*** (0.745 - 0.880) | **0.820** (0.746 - 0.884) |
| | MM$_{\text{Proj}}$ | 0.756 (0.688 - 0.819) | 0.758 (0.689 - 0.820) |

Table 8: Performance comparison of different missing value strategies for the DL-baseline (MM$_{\text{LR}}$) on Test$_{\text{complete}}$ using C-index, AUC, and their respective 95 % confidence intervals (5-year time horizon).

| Model | Modality | Imputation | ↑ C-Index | ↑ AUC |
|---|---|---|---|---|
| DL-Baseline | MM$_{\text{LR}}$ | -1 | 0.729 (0.656 - 0.794) | 0.733 (0.659 - 0.802) |
| | MM$_{\text{LR}}$ | Mean | 0.739 (0.659 - 0.813) | 0.745 (0.664 - 0.817) |
| | MM$_{\text{LR}}$ | Median | 0.732 (0.649 - 0.809) | 0.738 (0.654 - 0.812) |
| | MM$_{\text{LR}}$ | Embedding | 0.733 (0.655 - 0.805) | 0.738 (0.657 - 0.805) |

Table 9: Performance comparison of different missing value strategies for the DL-baseline (MM$_{\text{LR}}$) on Test$_{\text{complete}}$ using C-index, AUC, and their respective 95 % confidence intervals (10-year time horizon).

| Model | Modality | Imputation | ↑ C-Index | ↑ AUC |
|---|---|---|---|---|
| DL-Baseline | MM$_{\text{LR}}$ | -1 | 0.741 (0.702 - 0.780) | 0.765 (0.722 - 0.804) |
| | MM$_{\text{LR}}$ | Mean | 0.739 (0.694 - 0.781) | 0.758 (0.714 - 0.799) |
| | MM$_{\text{LR}}$ | Median | 0.753 (0.707 - 0.796) | 0.777 (0.731 - 0.820) |
| | MM$_{\text{LR}}$ | Embedding | 0.749 (0.705 - 0.790) | 0.772 (0.727 - 0.814) |

Table 10: Performance comparison of different missing value strategies for the DL-baseline (MM$_{\text{LR}}$) on Test$_{\text{reduced}}$ using C-index, AUC, and their respective 95 % confidence intervals (5-year time horizon).

| Model | Modality | Imputation | ↑ C-Index | ↑ AUC |
|---|---|---|---|---|
| DL-Baseline | MM$_{\text{LR}}$ | -1 | 0.721 (0.636 - 0.798) | 0.727 (0.641 - 0.802) |
| | MM$_{\text{LR}}$ | Mean | 0.739 (0.672 - 0.804) | 0.744 (0.670 - 0.808) |
| | MM$_{\text{LR}}$ | Median | 0.746 (0.678 - 0.809) | 0.751 (0.680 - 0.813) |
| | MM$_{\text{LR}}$ | Embedding | 0.744 (0.678 - 0.806) | 0.748 (0.678 - 0.810) |

Table 11: Performance comparison of different missing value strategies for the DL-baseline (MM$_{\text{LR}}$) on Test$_{\text{reduced}}$ using C-index, AUC, and their respective 95 % confidence intervals (10-year time horizon).

| Model | Modality | Imputation | ↑ C-Index | ↑ AUC |
|---|---|---|---|---|
| DL-Baseline | MM$_{\text{LR}}$ | -1 | 0.752 (0.706 - 0.796) | 0.776 (0.731 - 0.818) |
| | MM$_{\text{LR}}$ | Mean | 0.722 (0.683 - 0.763) | 0.741 (0.700 - 0.781) |
| | MM$_{\text{LR}}$ | Median | 0.740 (0.699 - 0.779) | 0.763 (0.721 - 0.803) |
| | MM$_{\text{LR}}$ | Embedding | 0.734 (0.696 - 0.775) | 0.756 (0.715 - 0.796) |

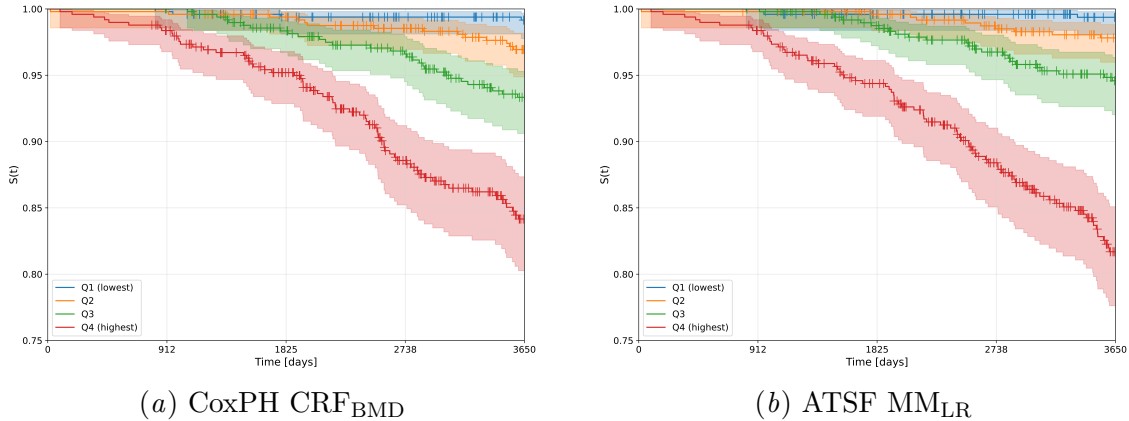

$(a)$ CoxPH CRF$_{\text{BMD}}$        $(b)$ ATSF MM$_{\text{LR}}$

Figure 4: Kaplan–Meier curves for CoxPH CRF$_{\text{BMD}}$ $(a)$ and ATSF MM$_{\text{LR}}$ $(b)$ over a 10-year time horizon on Test$_{\text{reduced}}$. Participants were stratified into quartiles based on each model's predicted risk (Q1 = lowest risk, Q4 = highest risk). Less opaque areas denote 95 % confidence intervals.

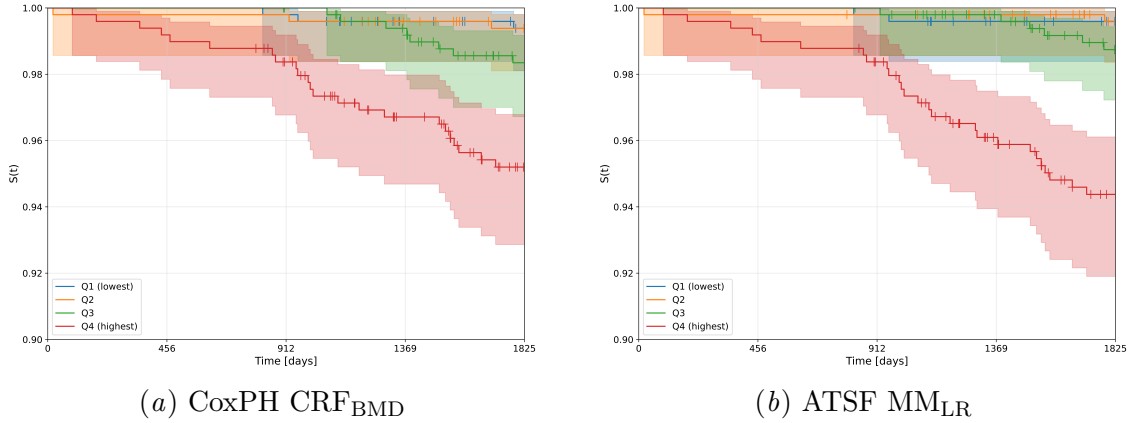

$(a)$ CoxPH CRF$_{\text{BMD}}$        $(b)$ ATSF MM$_{\text{LR}}$

Figure 5: Kaplan–Meier curves for CoxPH CRF$_{\text{BMD}}$ $(a)$ and ATSF MM$_{\text{LR}}$ $(b)$ over a 5-year time horizon on Test$_{\text{reduced}}$. Participants were stratified into quartiles based on each model's predicted risk (Q1 = lowest risk, Q4 = highest risk). Less opaque areas denote 95 % confidence intervals.

