# OpenReview forum: "Attention-to-Survival: Multimodal Fracture Risk Prediction Based on Pelvic Radiographs and Clinical Data from the Study of Osteoporotic Fractures"
_MIDL.io/2026/Conference — MIDL 2026 Poster_

### Official Review · Reviewer_djke · 2025-12-31

**Confidence:** 3
**Preliminary Rating:** 4
**Final Rating:** 4

**Summary:**

This paper introduces Attention-to-Survival Fusion (ATSF), a multimodal framework for hip fracture risk prediction that integrates pelvic radiographs with clinical factors,. By employing attention-based fusion with a programmable token and a DCTM head, the model effectively handles missing variables and time-dependent outcomes,. Evaluations on a large longitudinal cohort show that ATSF significantly outperforms FRAX and deep learning baselines in C-index and AUC, while providing interpretable attention maps.

**Strengths:**

- By integrating radiographic features and clinical risk factors (CRFs) through an attention-based fusion mechanism and a DCTM survival head, ATSF achieves strong predictive performance, with a reported C-index of 0.805 under bilateral imaging, and outperforms other baselines in the presented experiments.
- The framework relies on widely available pelvic X-rays rather than DXA scans and explicitly supports missing clinical variables. Experimental results suggest that performance remains relatively stable when some CRFs are unavailable.
- The use of attention allows the model to produce visualizations highlighting anatomically relevant regions (e.g., femoral neck, greater trochanter) and to quantify the relative contribution of individual CRFs such as age and fracture history, which may aid clinical interpretation.

**Weaknesses:**

- The cohort spans more than two decades, during which osteoporosis management and preventive treatments have changed substantially. Treatment history, a major confounder that directly affects fracture risk and survival time, is not included as a clinical risk factor, which may bias risk estimation.
- ATSF combines multiple complex components (ViT, Q-Former, multi-layer fusion, and a DCTM survival head), but the paper does not systematically compare alternative survival heads or simpler multimodal baselines within the same framework. This makes it difficult to isolate which components are essential to the observed gains.
- While the paper positions pelvic radiographs as a more accessible alternative to DXA, it does not evaluate an ATSF variant that includes BMD as a clinical risk factor. This leaves open whether the proposed imaging features provide incremental value beyond the current clinical gold standard, or whether their effects are largely redundant.

**Detailed Comments:**

- To process full-pelvis radiographs, the authors rescale MedVAE-derived projections and accept minor geometric distortion. A brief discussion or analysis of how this rescaling affects the extraction of fine trabecular bone features would strengthen the methodological clarity.
- Images for which the keypoint-based detection algorithm fails are excluded from analysis. Reporting the failure rate would help assess the robustness of the overall pipeline in real-world clinical settings with variable image quality.
- Additional ablation results isolating the contributions of major components (e.g., ViT vs. simpler encoders, Q-Former compression, programmable token vs. standard pooling) would help clarify which design choices are most critical.

**Justification Of Final Rating:**

The authors provide helpful clarifications in the rebuttal that address some of my questions and improve the clarity of several methodological aspects. I will keep my overall assessment and original score.

**Justification Of The Preliminary Rating:**

The paper presents a multimodal survival framework that integrates radiographic imaging and clinical risk factors and shows improved performance over established clinical baselines. The model uses attention-based fusion to combine modalities, supports missing clinical data, and provides anatomically informed attention visualizations. However, limitations related to population generalizability, attribution of performance gains across model components, and the handling of key clinical confounders remain insufficiently explored.

**Questions To Address In The Rebuttal:**

- The bilateral hip configuration consistently outperforms the full-pelvis projection. Could the authors clarify whether this gap is mainly due to information loss introduced by the MedVAE projection, or because high-resolution hip crops better preserve delicate trabecular structural details essential for risk prediction?
- ATSF is designed to handle missing CRFs, but $FRAX^{®}$ and $CoxPH$ are evaluated only on fully observed $Test_{reduced}$ cases. Could the authors provide a comparison with a CoxPH model using standard handling of missing covariates on the full $Test_{complete}$ set to better isolate the benefit of ATSF’s architectural handling of missingness?
- The model performs better at shorter horizons (e.g., 5-year) than at longer horizons (10-year). Do the authors interpret this as radiographic features capturing relatively transient structural risk that lose predictive relevance over time due to continuous bone remodeling, or are there other factors driving this difference?
- Images with failed keypoint detection are excluded from the study. Could the authors report the failure rate and describe the characteristics of these excluded cases? Is there a risk that patients with severely degenerated or anatomically distorted hips, who may be at higher fracture risk, were systematically excluded, and how might this affect the model’s reliability in real-world clinical settings?

---

> ### Author Response · Authors · 2026-01-23
> **Detailed Comment  to Reviewer djke**
>
> ## Weaknesses:
>
> Disregard of treatment history
>
> - We appreciate this insightful comment. Treatment history is indeed an important factor that could influence fracture risk and survival time. Unfortunately, this information was not included in the analysis plan when requesting the SOF dataset, and thus was not available for our study. Incorporating treatment history would be an interesting and valuable extension for future research to further refine risk estimation.
>
> Ablation study not considering each component individually
>
> - We agree that a more extensive ablation study and comparison against alternative survival heads or simpler multimodal baselines would provide additional insight into the contributions of each component. While such analyses are certainly valuable, they were not feasible within the scope of the current study due to computational and resource constraints. We consider this a promising direction for future work.
>
> Disregard of DXA BMD
>
> - The exclusion of BMD as a clinical risk factor was a deliberate choice, guided by the clinical motivation of this study to focus on widely available pelvic radiographs as an alternative to DXA imaging. Including BMD would have conflicted with this objective. Nonetheless, evaluating a hybrid model that combines pelvic radiographs with BMD could be an interesting avenue for future research.
>
>
> ## Detailed Comments:
>
> Effect of MedVAE and rescaling on trabecular bone structures
>
> - The assessment of rescaled MedVAE-derived projections is largely based on the findings reported in the MedVAE paper, and we also performed qualitative checks to ensure that relevant features were preserved.
>
> Failure rate of keypoint detection
>
> - The keypoint-based detection algorithm achieves an accuracy of 94% for the left hip and 96% for the right hip. Images for which keypoint detection fails are excluded from the analysis. In these excluded images, the proportion of fracture cases is 14.4% for the left hip (72 out of 501) and 13.6% for the right hip (45 out of 331), which is similar to the overall fracture ratio in the dataset. Therefore, the exclusion of these cases does not disproportionately remove fracture cases, and the robustness of the pipeline is maintained. We reported these numbers in the revised manuscript to clarify the impact of keypoint detection failures.
>
> Additional ablation studies
>
> - As noted previously, a more extensive ablation isolating the contributions of major components (e.g., ViT versus simpler encoders, Q-Former compression, programmable token versus standard pooling) would indeed provide valuable insights that we would like to address as future work.
>
>
> ## Questions to Address in the Rebuttal:
>
> Effect of MedVAE and rescaling on trabecular bone structures
>
> - We suspect that the slightly lower performance of the full-pelvis projection compared to the bilateral hip configuration is primarily due to information loss introduced by the MedVAE projection. Contrary to the results reported in the MedVAE paper, it appears that not all trabecular structures are fully preserved, and these structures may have a meaningful impact on prediction accuracy. Additionally, the full-pelvis images include markers and other regions that may introduce confounding signals, whereas the bilateral hip (LR) configuration focuses on the relevant regions.
>
> Performance comparison using standard handling of missing values
>
> - Thank you for the comment. We implemented standard strategies for missing values and reported the results in the revised manuscript.
>
> Discussion on the influence of the time horizon on performance
>
> - We interpret the higher performance at shorter horizons, such as 5 years, as partly reflecting that radiographic features capture structural risk factors whose predictive relevance may decrease over time due to ongoing bone remodeling. In addition, some CRFs are time-varying; for instance, participants may start or stop smoking during the follow-up period. The occurrence of these changing factors is naturally lower over shorter horizons, which may also contribute to the observed differences in model performance between 5- and 10-year predictions. A description of these time-varying features is provided in Appendix A.
>
> Systematic errors caused by removing samples with incorrect keypoints
>
> - As previously mentioned, the keypoint-based detection algorithm achieves an accuracy of 94% for the left hip and 96% for the right hip, and we reported the corresponding failure rates in the revised manuscript. While there is always a potential risk that patients with severely degenerated or anatomically distorted hips (who may be at higher fracture risk) could be excluded, the relatively small number of such cases suggests that this effect is likely limited. Importantly, the model’s predictions are not biased by the presence of implants or other image artifacts, which we consider a more relevant factor for robustness in real-world clinical settings.

---

### Official Review · Reviewer_6gRv · 2026-01-09

**Confidence:** 5
**Preliminary Rating:** 4
**Final Rating:** 4

**Summary:**

This paper proposes an attention-based conditional transformation framework for survival prediction in osteoporotic fracture risk, integrating pelvic radiograph features with clinical risk factors within a Cox proportional hazards model. The approach is evaluated on a large cohort of 7,825 women followed for up to 23 years, demonstrating that multimodal fusion outperforms clinical-only and image-only baselines in terms of C-index and AUC. The overall topic is important and clinically relevant, and the multimodal modeling direction is promising. The potential can be seen beneficial to enhance fracture risk screening.

**Strengths:**

•	Important problem from a clinical aspect (osteoporotic fracture risk and survival)

•	Large longitudinal cohort with multimodal data (radiographs + clinical risk factors)

•	Reasonable performance gains from multimodal modeling

•	Methodologically sound core idea (an attention-based conditional transformation)

**Weaknesses:**

•	Clarity in the related work and contribution

•	Limited justification of experimental design choices (data split, evaluation strategy)

•	Missing result comparisons wrt state-of-the-art survival models cited in the paper

•	Survival curve analysis and deeper discussion of time horizons and missing data effects

**Detailed Comments:**

This paper proposes a conditional transformation model with an attention mechanism to predict fracture-related survival outcomes using a Cox proportional hazards (CoxPH) framework in aging populations affected by osteoporotic fractures. The study analyzes a large longitudinal cohort of 7,825 women followed over a 23-year period and integrates pelvic radiograph features with clinical risk factors (CRFs). The CRFs include nine variables: sex, age, BMI, smoking status, alcohol consumption, prior fractures, rheumatoid arthritis, glucocorticoid exposure, and parental history of hip fracture.

The overall topic is important and clinically relevant, and the multimodal modeling direction is promising. However, several aspects of the manuscript require clarification, in addition to a more comprehensive evaluation to clarify the importance of the topic and its effect.

Section 2: Related Work

- The authors provide a broad overview of survival modeling approaches in the literature, however, the review emphasizes limitations of prior studies without in-depth discussion on their performance or strengths. A more balanced discussion of reported results would help clarify the effect of the proposed method.

- The manuscript does not clearly specify specific gaps in existing work nor explicitly state how the proposed methodology addresses these gaps. A clear view of the contributions would be helpful to understanding the prior works standing versus multimodal survival models, particularly those that integrate imaging and clinical data.

Section 3.1: Data

- The authors state that the dataset originates from the NIH-funded Study of Osteoporotic Fractures and includes data collected across multiple centers. However, a clarification of the below points would help readers understand the data and limitations:

-- Was there a one-to-one pairing between clinical data and pelvic radiographs for all subjects?

-- As the data is provided by multicenter institutes, please add how this affects this study and how was data heterogeneity handled (e.g., imaging protocols, acquisition quality)?

-- How were missing values managed in both the clinical and imaging data?

-- Did the authors apply any normalization strategies applied across centers?

Clarifying these points would help in assessing the reliability and reproducibility of the results.

Section 3.4: Training Details

- The authors randomly split the dataset into training, validation, and testing sets using a 50%-10%-40% ratio. The rationale for this split is unclear, especially with respect to the relatively large dataset size. Common practices typically allocate a larger proportion of data to training (such as 70-80%), often combined with k-fold cross-validation, and reserve a smaller held-out test set. The authors should clarify:

-- Why such a large portion of data was reserved for testing

-- Why k-fold cross-validation was not adopted

- It is stated that the complete test set contains missing values, necessitating the use of a reduced test subset for fair comparison. However, please add:

-- Whether missing values were also present in the training and validation sets

-- How missing values were handled during model training

Although the distinction between Test reduced and Test complete is reasonable, explanation of this experimental design and justification of it would be helpful in understanding the limitations and requirements of the implementation.

Section 3.5: Evaluation and Ablation

- The authors apply bootstrapping on the test set for uncertainty understanding and compute 95% confidence intervals. This is a valid approach, however, the manuscript does not explain why strategies such as k-fold cross-validation were not used, especially during training and model selection.

- Although the authors report the concordance index (C-index) and AUC to assess predictive accuracy and discrimination, there is no mention of Kaplan-Meier survival curves, which is notable. Kaplan-Meier analysis could provide valuable insight into risk stratification and demonstrate whether the model meaningfully separates high-risk and low-risk patient groups.

Section 4: Results

- Table 1 shows that the clinical baselines achieve a C-index of 0.769 and AUC of 0.788. Image-only models (unilateral and bilateral) achieve lower performance, whilst the multimodal model combining bilateral imaging and clinical factors achieves the best performance. While the results are promising, the authors do not compare their method against state-of-the-art survival models discussed in the related work. Given that these methods are explicitly mentioned, their omission from experimental comparisons weakens the validation.

- The choice of CoxPH as the primary survival backbone requires stronger justification. The authors should clarify whether alternative survival models (such as DeepSurv, DeepHit, or Random Survival Forests which are mentioned in the prior work) were explored and, if not, why CoxPH was preferred over more expressive nonlinear approaches.

Section 6: Conclusion, Limitations, and Future Work

The manuscript would benefit from a discussion on:
-- How excluding BMD may impact predictive performance.
-- Whether future work could explore hybrid settings where BMD is available.
-- How this limitation affects generalization and clinical adoption.

Additional Results

The manuscript reports results for:
-- test reduced at a 5- and 10-year horizon.
-- test complete at 5- and 10-year horizons.

- The authors should justify the clinical significance of the 5- and 10-year horizons and explain why other time points were not considered.

- The results indicate limited improvement and in some cases a slight performance drop when moving from test reduced to test complete at the 5-year horizon. Specifically, the multimodal model achieves and c-index of 0.805, AUC of 0.826 in the test reduced and a c-index 0.800 and AUC of 0.821 in test complete. The authors should discuss potential reasons for this drop and explain why performance does not improve when using the larger test set.

**Justification Of Final Rating:**

The authors have made some constructive revisions and provided detailed responses which have improved the clarity and presentation of the paper. The addition of KM-curves while beneficial shows some overlap between different risk groups. A stronger analysis would have included two group risk stratification to demonstrate the overlap between high risk and low risk groups. The model is still reliant on a linear Cox PH model on top of deep representations which limit the interpretability of the model and the absence of comparison with more expressive survival models and cross dataset generalization remains as limitations of the study. Overall as this work is solid with relevant contributions but its experimental scope and methodology depth are limited, my final rating will remain the same as before.

**Justification Of The Preliminary Rating:**

The paper addresses a clinically meaningful problem with a large cohort which makes it interesting from a biomedical aspect. The multimodal survival framework is interesting and solid, however, more input should be provided to clarify some methodological and experimental aspects of the paper.

**Questions To Address In The Rebuttal:**

Mostly the results from different aspects including the Test_reduced vs Test_complete, performance vs missing CRFs and whether the Kaplan-meier curves can be added as they help identify high-risk patients.

---

> ### Author Response · Authors · 2026-01-23
> **Detailed Comment  to Reviewer 6gRv (3/3)**
>
> ## Results:
>
> Comparison against SOTA survival models mentioned in related work
>
> - A direct comparison with some of the state-of-the-art survival models mentioned in the related work is not possible, as these approaches perform classification at a predefined event time rather than true survival analysis. Therefore, we have focused our experimental comparisons on the DL baseline inspired by Kong et al., as well as established clinical baselines such as FRAX and CoxPH. This approach allows for a meaningful evaluation of model performance in the context of time-to-event prediction, which is the focus of our work.
>
> Justification of CoxPH as primary survival backbone
>
> - We chose CoxPH as the primary survival backbone because it is a well-established and widely used method in clinical research, providing a strong and interpretable baseline. We agree that exploring alternative survival models, such as DeepSurv, DeepHit, or Random Survival Forests, could be interesting for future work. While these more expressive nonlinear approaches may offer potential benefits, evaluating them was beyond the scope of the current study, which primarily focuses on multimodal integration.
>
>
> ## Conclusion, Limitations, and Future Work:
>
> Impact of DXA on performance
>
> - We agree that including BMD could potentially impact predictive performance and consider this an interesting avenue for future work. However, incorporating BMD data would not support the clinical motivation of our study, which focuses on widely available pelvic radiographs and clinical risk factors.
>
> Hybrid setting including BMD when available
>
> - Exploring hybrid settings where BMD is available is indeed an interesting direction for future work.
>
> Effect on generalization and clinical adoption
>
> - Our motivation was to only rely on widely available radiographs and CRFs as this may increase practical applicability. Incorporating a hybrid setting mentioned before with BMD where available could be an optional extension to further support clinical adoption, but this was not the focus of the current study.
>
>
> ## Additional Results:
>
> Justification of time horizons selected
>
> - The choice of 5- and 10-year horizons was guided by existing clinical guidelines and baseline models. The 10-year horizon aligns with the FRAX assessment period, allowing for a meaningful comparison to this widely used clinical tool. The 5-year horizon was selected to remain close to guideline recommendations (3 years according to the Dachverband Osteologie (DVO), see https://leitlinien.dv-osteologie.org/wp-content/uploads/2024/02/Kurzfassung-verlinkt-mit-Langfassung-der-DVO-Leitlinie-DVO-2023-V2.1.-ROT-003.pdf) while still ensuring a sufficient number of positive cases in the test set to obtain stable and statistically meaningful performance estimates.
>
> Reasons for performance differences between test sets
>
> - The slight drop in performance from the reduced to the complete test set at the 5-year horizon may be related to the presence of missing values in some CRFs. For instance, if highly influential features such as age are missing for certain participants, predictions for these cases could be less accurate, which may contribute to the observed differences in average performance metrics. While we cannot definitively quantify this effect, this explanation is consistent with our feature importance analysis and the design of the test sets.

---

> ### Author Response · Authors · 2026-01-24
> **Detailed Comment  to Reviewer 6gRv (2/3)**
>
> ## Training Details:
>
> Justification of data splitting
>
> -  The relatively large test split was chosen to ensure a sufficient number of positive fracture events for a statistically meaningful evaluation. Given the high censoring rate, smaller test sets would have resulted in unstable and high-variance estimates of the evaluation metrics, potentially leading to misleading conclusions about model performance. We therefore prioritized a robust assessment on a large held-out test set over marginal gains in training set size.
>
> Justification for not using k-fold cross-validation
>
> - K-fold cross-validation was not used in this study, primarily due to computational constraints and the use of predefined train, validation, and test splits. Given the high censoring in the survival data, folds could potentially show variable event distributions, which might increase variance in the evaluation metrics. Moreover, the relatively large test set already provides a sufficient number of positive events, allowing for a robust assessment of model performance.
>
> Missing values in training and validation sets
>
> - Yes, missing values were also present in the training and validation sets. This did not pose a problem for FRAX (all values were provided in the SOF dataset) and was only relevant for the DL baseline and ATSF models. For the CoxPH models, participants with missing values in the relevant variables were excluded. We clarified this in the manuscript.
>
> Handling of missing values during training
>
> - Missing values during model training were handled as described earlier. Briefly, all participants had pelvic radiographs, and missing values occurred only in CRFs. In the DL baseline, missing CRF values were encoded as −1. In ATSF, we implemented both a learnable missing token and a masking strategy, with the final reported results using the masking approach to exclude missing values from learning.
>
> Limitation and requirements of implementation based on test sets?
>
> - We created the reduced and complete test sets to enable a fair and differentiated comparison between models. The reduced test set ensures that all models are evaluated on the same subjects, allowing for a controlled comparison. The complete test set, which includes participants with missing values in CRFs, is used to assess how the models handle missing values and whether performance remains stable, decreases, or improves when these extra samples (with missing values) are included. We clarified this experimental design and its rationale in the revised manuscript.
>
>
> ## Evaluation and Ablation:
>
> Justification for not using k-fold cross-validation (e.g., for model selection)
>
> - We agree that k-fold cross-validation could have provided additional insight and might be helpful in some contexts. In our study, we opted for predefined train, validation, and test splits to allow consistent model selection and evaluation. Given the complexity of the ATSF model, performing full k-fold cross-validation would have required more computational resources. We noted this as a limitation in the revised manuscript, while emphasizing that the large test set and bootstrapping procedure provide robust estimates of performance uncertainty.
>
> Kaplan-Meier Analysis
>
> - Thank you for the comment. We have included Kaplan-Meier survival curves in the appendix. For this analysis, participants were stratified into four risk quartiles based on the model’s predicted risk scores. The resulting curves show a clear separation between groups, indicating that the model is able to distinguish patients with different levels of fracture risk.

---

> ### Author Response · Authors · 2026-01-24
> **Detailed Comment  to Reviewer 6gRv (1/3)**
>
> ## Related Work:
>
> More balanced discussion of related work
>
> - Thank you for this comment. We revised the related-work section accordingly to better contextualize existing approaches and to more clearly delineate the contribution of the proposed method.
>
> Specification of gaps and how they are addressed
>
> - The manuscript already discusses specific gaps in existing work and positions the proposed methodology accordingly. We distinguish between approaches that perform classification at predefined time points rather than survival analysis in the strict sense, methods relying on two-stage pipelines, and studies using different data modalities. We have further refined the discussion of related work to more explicitly highlight how our proposed approach addresses these gaps and why it is needed in the context of existing work.
>
>
> ## Data:
>
> One-to-one pairing of clinical and imaging data
>
> - Yes, there is a one-to-one pairing between clinical data and pelvic radiographs for all subjects included in the present study. As noted in the manuscript, some participants have missing values in certain CRFs; however, the corresponding pelvic radiograph is available for every subject.
>
> Handling of data from multiple clinical sites
>
> - The multicenter nature of the SOF dataset was considered in the analysis. While handling center-specific heterogeneity was not the primary focus of this study, we examined the data distribution across participating institutions and observed comparable distributions without pronounced center-specific biases. Based on this assessment, no additional harmonization or center-specific normalization methods were used. We clarified this point in the manuscript to better contextualize potential limitations related to data heterogeneity.
>
> Handling of missing data
>
> - For all included participants, a pelvic radiograph was available; therefore, no missing values occurred in the imaging data. Missing values were present only in the CRFs.
> To enable a differentiated analysis, we deliberately constructed two different test sets (with/without missing values). In the DL baseline, missing CRF values were encoded as −1, as the original work from Kong et al. does not explicitly describe a missing-value handling strategy. In ATSF, we implemented two strategies: first, a learnable missing token that allows missing values to be incorporated into the cross-attention mechanism, effectively treating missingness as potentially informative; second, a masking-based approach in which missing values are explicitly masked and excluded from learning. As discussed in the paper, we ultimately report results using the masking strategy, as it better reflects the assumption that missingness is non-informative. We clarified this handling of missing values in the revised manuscript.
>
> Normalization strategies for data from multiple clinical sites
>
> - No center-specific normalization strategies were applied. As mentioned previously, an analysis of the data distribution across participating centers indicated comparable distributions without pronounced center-dependent differences. Based on this observation, additional cross-center normalization was not deemed necessary.

---

> > ### Comment · Reviewer_6gRv · 2026-01-25
> >
> > For the related work, can the authors list the most closely related multimodal survival models they discussed and briefly summarize (1–2 sentences) how their method differs in methodology?
> >
> > Concerning the missing CRF values, you mentioned that they were encoded as -1. Can you elaborate on whether these values might bias the baseline performance and whether the use of alternative approaches, such as mean, median, or even learned embeddings, changes comparative results?

---

> > > ### Author Response · Authors · 2026-01-29
> > > **Response to Reviewer 6gRv (1/2)**
> > >
> > > ### For the related work, can the authors list the most closely related multimodal survival models they discussed and briefly summarize (1–2 sentences) how their method differs in methodology?
> > >
> > > - Thank you for your interest in our method. The most closely related model to our work is the one by Kong et al. (2022). Our proposed ATSF model differs from this baseline in the fusion of multimodal inputs (radiograph, CRFs). In the study by Kong et al., the combination of radiographic features and CRFs is achieved by implicit feature concatenation, followed by an MLP. In contrast, our ATSF employs an attention-based fusion mechanism with a task-specific query token that dynamically aggregates the most relevant information from both modalities. More specifically, we use cross-attention (an approach for learning relationships across the used modalities) which – as indicated by our experiments – yields a more meaningful latent representation of fracture risk.
> > > Another approach for survival fracture risk was proposed by Shaik et al. (2024), which follows a two-stage process. In the first stage, a CNN model is trained to classify whether a patient will suffer a fracture within the next 10 years. In the second stage, the CNN output score is combined with a small set of CRFs and then used as input for a CoxPH model. As a result, this approach does not learn common representations or interactions between image features and CRFs and remains constrained by the proportional hazards assumption of the CoxPH model. We discussed these differences in the revised version of our draft (highlighted in red).

---

> > > > ### Author Response · Authors · 2026-01-29
> > > > **Response to Reviewer 6gRv (2/2)**
> > > >
> > > > ### Concerning the missing CRF values, you mentioned that they were encoded as -1. Can you elaborate on whether these values might bias the baseline performance and whether the use of alternative approaches, such as mean, median, or even learned embeddings, changes comparative results?
> > > >
> > > > - Thank you for pointing this out. We agree that encoding missing values as −1 could potentially introduce a small bias, as it assigns a specific numerical value to missing values. To assess whether this affects the baseline performance, we conducted additional experiments.
> > > > As suggested, we retrained our best DL- baseline ($\text{MM}_\text{LR}$) using three alternative strategies for handling missing values: mean imputation, median imputation, and an embedding-based approach. All variants were evaluated on both test sets (Test Complete and Test Reduced) and for both time horizons (5 and 10 years).
> > > > For the embedding-based approach, embeddings were applied only to categorical variables. Missing categorical values were treated as a separate category and mapped to a dedicated embedding vector. Continuous variables were not embedded, as embedding all real-valued inputs would require an infinitely large codebook. Instead, missing continuous variables were imputed using the mean, ensuring comparability with the mean-imputation baseline.
> > > > For mean and median imputation, these strategies were applied only to continuous variables. Applying mean or median imputation to categorical variables would produce intermediate values between 0 and 1, which do not have a clear semantic interpretation. Therefore, missing categorical variables were consistently set to 0 across all strategies, ensuring a fair and aligned comparison.
> > > > Across all experiments, the results were consistent (see appendix in revised manuscript). None of the strategies resulted in statistically significant performance differences compared to the original setup, indicating that the potential bias introduced by encoding missing values as −1 is small.
> > > > Nevertheless, some trends can be observed. Median imputation was most often the best or second-best performing strategy across both test sets and time horizons. Mean imputation showed very similar performance, typically slightly below median imputation. The embedding-based approach generally performed similarly to mean imputation but did not provide consistent improvements. Similarly, the -1 encoding showed comparable performance to mean and median imputation, with only minor differences.
> > > > The limited impact of the embedding-based approach can be explained by the categorical variables. In this setting, missing categorical values are effectively binary (observed vs. missing). As a result, the embedding layer mainly learns a representation for missing values, which is functionally similar to using a fixed placeholder value combined with learned biases in later layers. Since the handling of continuous variables remains unchanged, the overall influence of embeddings on model performance is limited.
> > > > Overall, these findings suggest that while different strategies for handling missing values may introduce small variations in performance, they do not significantly affect the main conclusions of our study. Among the tested approaches, median imputation appears to be a simple and robust alternative for handling missing continuous CRF values. We will add the results to the appendix and write a sentence about different imputation strategies in Section 3.2.

---

### Official Review · Reviewer_QArX · 2026-01-09

**Confidence:** 4
**Preliminary Rating:** 3
**Final Rating:** 4

**Summary:**

This work proposes an end-to-end deep survival analysis model for time-to-hip-fracture prognosis from pelvic radiographs and clinical metadata. Compared to previous work, it adds flexible input data handling (any non-informative censoring, missing data, and multiple modalities) and approximate attributions of input image regions and clinical variables. The authors developed a (vision) transformer based architecture and combine it with a deep survival analysis head (DCTM), termed ATSF. The model is compared against an established fracture risk assessment tool, the classical linear regression model CoxPH, and an adjusted two-stage deep learning based fracture prognosis model from the literature. While the risk tool is based on clinical metadata only, the CoxPH could alternatively accept DXA-derived bone mineral density features, and the deep learning based model allowed for DXA images as inputs in addition to clinical metadata. The proposed method, on the other hand, uses pelvic radiographs and clinical metadata as inputs and outperforms the baselines for the prognosis of hip fracture within 5 and 10 years wrt. Harrell's C-index and AUC for binary outcomes.

**Strengths:**

The manuscript is very well written with a good level of detail. The addressed research gap, its fit to MIDL, and the choice in model architecture design, are well grounded and explained. The comparison baselines seem reasonable and the authors follow a recommended approach for statistical model comparison suited for their setup. Sensible ablations were added, importantly, one that supports the claim of good missing data handling. The work successfully shows that a), pelvic radiographs and clinical variables can be used to predict hip-bone fracture risk, and that b), their developed model using these inputs has better discriminative performance than previous work which uses different inputs, while adding input handling flexibility. The topic of survival analysis is introduced in the appendix, allowing readers a quick access to the context of the present work.

**Weaknesses:**

While overall, the baselines are good choices, some changes and clarifications are needed regarding the DL-baseline, which appears rather weak. Further, the authors use two metrics that are biased in the presence of the high censoring rate. Additionally, some technical details are missing.

**Detailed Comments:**

- Baselines
	- Re: "Shaikh et al. (2024) employ a two-stage pipeline that omits relevant CRFs and inherits the assumptions of Cox proportional hazards model (CoxPH). In contrast, Kong et al. (2022), while conceptually closest to our approach, focus exclusively on spinal radiographs and use a simple ResNet-based architecture [...]":
	The differentiation between Shaikh and Kong, as well as between Kong and the proposed work, could be improved. Kong et al. extract image features using a pretrained Resnet and then train DeepSurv on these with CRFs, making it a two-stage pipeline, too, and it inherits the proportional hazards limitation of CoxPH as well. While Kong et al. seems like a valid related work, and with that, a valid baseline, its encoder is not finetuned (and that alone might explain lower performance), which requires to at least acknowledge this difference clearly.
	- Furthermore, you exchange the DeepSurv, their core contribution, with an MLP + DCTM model to guide the focus on the multimodal data integration component of the models. Since the DCTM is more complex than the DeepSurv, with the latter being a special case of DCTM, this exchange would require more hyperparameter tuning than the DeepSurv, potentially lowering the performance if not tuned. However, it sounds as if you tuned only the DCTM of your proposed model, but not the DL-baseline's head. In that case, either a HP search would be required for the DL-baseline with DCTM head, or the use of the Kong et al. design, i.e., a DeepSurv model, presumably with the default settings of R-deepsurv, which the original work uses.
	- I would appreciate if the usage of DXA vs. pelvic images in DL-baseline vs. ATSF could be noted or highlighted in the discussion part like "...the multimodal bilateral DL baseline, which was trained on DXA imaging and CRFs, is better than all other baselines".
	- While I appreciate the depth of the SOF overview in Tab. 2, I did not understand if it contains DXA and pelvic radiographs for the same patients, and importantly, if these modalities were present for all patients in the reduced test set such that the evaluated subjects are identical for DL-baseline and ATSF.
	- Could you clarify if the DXA images had been (keypoint) aligned, since you give the pelvic radiographs this treatment and Kong et al. report alignment, too.
	- What exactly do you mean when saying that the DL-baseline received CRFs as binary indicators?
	- Since your goal is to show that pelvic radiographs are valid inputs and that your architecture can add multimodality and missing data handling, I understand that it is not strictly a necessity to add a stronger DL-baseline. If you, however, wanted to support performance claims further, you could add an end-to-end variant of Kong et al., where the CNN is trained using the DeepSurv loss.

- Architecture and training
	- I presume that all image-based models receive a snapshot and not a longitude of image inputs, however, an explicit statement about this is missing.
	- For how many epochs was the ATSF trained and how was the best epoch/best weights selected for ATSF and DL-baseline?
	- How do you justify the use of different augmentations derived from hyperparameter searches for different models? Shouldn't the augmentations be kept constant for all models, at least for those that share the same main architecture?
	- You use two discrimination metrics, C-index (Harrell) and AUC. The first is popular for survival analysis but known to be prone to bias in the presence of many censored subjects, and your reduced test set has a censoring rate of 85-94%. Consider using less biased metrics like the AUC(c,d)/C-index(Uno) alternatively or additionally. The second, the AUC for binary outcomes, cannot handle censoring and forces you to evaluate it in "classification style", as you noted. That is okay, but due to the apparent flaws of this setup, I consider it valid only as a secondary metric, next to a C-index. Please consider indicating that this context makes it a secondary metric and that the AUC should be interpreted with care. You could also consider to add a model calibration metric to add insight (Brier score(ipcw)).
	- You do not compare the potential of DXA vs. pelvic radiographs for hip fracture prognosis and yet argue that it adds information. To me, one can conclude from the experiments that they "provide clinically valuable structural information" but not that it "leads to substantially improved fracture risk estimation". For the latter conclusion, you could re-train and tune your model and hyperparameters on DXA + CRFs and compare. Alternatively, one could relax the claim (-> ...can lead to substantially improved risk estimation) and add a note on the potential interaction of model architectures and inputs compared.

- Misc
	- Code availability: I encourage the authors to provide code and weights prior to acceptance, since many works remain without these, even when initially promised, which drastically limits the impact of the work.
	- Data availablity: A sentence about the SOF data availability is missing.
	- Statistical test: Did you perform exactly one test for ATSF(MM-LR) against CoxPH(BMD) regarding c-index, i.e., your best model against the next best, focusing on c-index, or were multiple tests done? If so, did you adjust the p/alpha for multiple comparisons?
	- "negative cases" in 3.5 should be "negatives" or alike to not confuse the reader with "cases" which by convention are "positives" in medical contexts.
	- "SurfTrace ([...] and Sun)": I like the painted figure here, but it is "SurvTrace"

**Justification Of Final Rating:**

The revision was substantial and it added clarity and detail to the manuscript. My points were addressed in the discussion phase and given that, I am happy to increase my rating to 4, being torn between 4 and 5.

**Justification Of The Preliminary Rating:**

3/Borderline was selected, because a few too many details are still lacking the manuscript in order to constitute a comprehensive well-grounded submission. However, with the comments addressed, this work will make a good MIDL paper.

**Questions To Address In The Rebuttal:**

All of the above questions should be commented on and/or incorporated.

---

> ### Author Response · Authors · 2026-01-23
> **Detailed Comment  to Reviewer QArX (3/3)**
>
> ## Misc:
>
> Code availability
>
> - We fully understand and appreciate the importance of code and weight availability. In our previous publications (e.g., [https://papers.miccai.org/miccai-2025/paper/3144_paper.pdf], [https://openreview.net/pdf?id=dgusajTAqF]), we have consistently made the full code publicly available. We are currently preparing the code and trained model weights for this work and will provide them as soon as possible.
>
> Data availability
>
> - Please note that the data is not publicly owned by us. We now added a direct link to the official SOF website [https://sofonline.ucsf.edu/].
>
> Statistical testing
>
> - Only a single statistical test was performed, comparing the ATSF (MM-LR) model to the next best baseline, CoxPH (BMD), with respect to the C-index.
>
> Stylistic errors/typos
>
> - We thank the reviewer for these helpful corrections which we addressed in the revised manuscript.

---

> ### Author Response · Authors · 2026-01-24
> **Detailed Comment  to Reviewer QArX (2/3)**
>
> ## Architecture and training:
>
> Clarification of temporal dimension of the inputs used
>
> - All image-based models receive only a single pelvic radiograph from the initial screening, rather than a longitudinal series of images. We explicitly stated this in the revised manuscript to avoid any ambiguity regarding the input data.
>
> Best model selection criterion and ATSF training epochs
>
> - The ATSF model was trained for 150 epochs. The selection of the best epoch and corresponding model weights for both ATSF and the DL baseline was based on the highest concordance index (C-index) achieved on the validation set. We included these training details in the revised manuscript to improve clarity and reproducibility.
>
> Different augmentations for same models through HPO
>
> - We agree that for inputs of the same modality, such as left-only (L) versus bilateral (LR) pelvic radiographs, it would have been reasonable to use the same augmentation parameters for models with the same main architecture. In our current experiments, this was not explicitly enforced. For inputs that are fundamentally different, such as the MedVAE projection, distinct augmentations were applied. While this was not systematically controlled across all models, we do not expect it to affect the overall conclusions regarding the impact of multimodal integration.
>
> AUC as secondary metric
>
> - We agree with this observation and clarified in the manuscript that the AUC should be interpreted with caution and serves as a secondary metric alongside the C-index.
>
> Pelvic radiographs vs. DXA
>
> - As previously noted, our study does not use DXA imaging, and therefore we cannot directly compare the predictive potential of DXA versus pelvic radiographs. We have toned down the corresponding claims in the manuscript to state that pelvic radiographs “can provide clinically valuable structural information and can lead to improved fracture risk estimation,” rather than implying a definitive performance improvement.

---

> ### Author Response · Authors · 2026-01-24
> **Detailed Comment  to Reviewer QArX (1/3)**
>
> ## Baselines:
>
> Differentiation between Shaikh et al., Kong et al., and our approach
>
> - Thank you for the comment. We agree that the differentiation between Shaikh et al. (2024), Kong et al. (2022), and our work should be clarified and made more precise in the paper.
> Following our definition, Shaikh et al. (2024) employ a two-stage pipeline and use CT data rather than radiographs. In contrast, we consider Kong et al. (2022) as a single-stage approach, as the pretrained ResNet is used as a feature extractor and the entire model is optimized jointly with a survival loss. We agree, however, that due to the use of DeepSurv, this approach inherits the proportional hazards assumption of CoxPH, which is explicitly stated. More details reflecting these distinctions more clearly can be found in the related-work section of the revised manuscript.
>
> Modification of DL-Baseline (other survival head)
>
> - Our intention in replacing the DeepSurv head was to focus the comparison on the multimodal data integration rather than on differences between survival heads. In our view, the core contribution of Kong et al. lies in the multimodal combination of structured clinical variables and imaging-derived features for fracture risk modeling, rather than in the specific choice of the survival head itself. Accordingly, the MLP architecture used for feature fusion remains identical, and only the prediction head was exchanged.
> We chose the DCTM head based on prior work demonstrating its strong performance across survival modeling tasks and to establish a more consistent comparison across the two multimodal models used in this study. Our goal was not to optimize for the best-performing survival head per se, but to analyze the impact of multimodal fusion under a unified survival modeling framework. We acknowledge that the DCTM is more flexible than DeepSurv and that this choice should be justified more clearly. We will therefore clarify in the revised manuscript that the DL baseline uses the same MLP architecture, that only the head differs, and we will explicitly discuss the implications of this design choice, including the scope of hyperparameter tuning.
> We have revised the paper accordingly to make this design decision and its motivation more transparent.
>
> DXA usage in DL-Baseline
>
> - Thank you for this comment. We would like to clarify that neither the DL baseline nor the proposed ATSF model uses DXA imaging or DXA-derived BMD values. Both models are trained exclusively on pelvic radiographs in combination with clinical risk factors, in line with the clinical motivation of our work to rely on widely available imaging data beyond DXA.
> We agree that this distinction should be made more explicit to avoid confusion. We have therefore revised the discussion section to clearly state the imaging modalities used by all baselines and by ATSF, and to emphasize that DXA data are intentionally not included.
>
> DXA images in SOF dataset
>
> - Thank you for this comment, to clarify: All patients included in the reduced test set have an available pelvic radiograph, which is consistently used across all image-based models, including both the DL-baseline and ATSF. Therefore, the evaluated subject cohorts are identical for these models.
> As noted previously, DXA imaging data and DXA-derived BMD are not used in any of the multimodal or image-only experiments. In the revised manuscript, we now report the availability of pelvic radiographs and the exclusion of DXA data more explicitly to avoid any potential confusion.
>
> DXA images keypoint alignment
>
> - As noted before, no DXA images were used in our study. The BMD information was available directly in the dataset and did not involve any imaging-based processing. Therefore, no keypoint alignment was performed for DXA, and all image preprocessing and alignment steps only apply to the pelvic radiographs.
>
> CRFs as binary indicators in the DL-Baseline
>
> - By stating that the DL baseline received CRFs as binary indicators, we mean that all categorical clinical risk factors were encoded as 0 or 1. Continuous variables were normalized to the [0, 1] range, and missing values were indicated with -1. All CRFs were combined into a single feature vector, which was then processed by the DL baseline alongside the image features.
> We clarified this encoding scheme in the manuscript to ensure the preprocessing of both categorical and continuous variables is clearly understood.
>
> End-to-end training of DL-Baseline
>
> - We agree with the reasoning provided and appreciate the suggestion. In our initial submission, we deliberately did not include an end-to-end variant of Kong et al., as our primary focus was not to optimize the baseline but to demonstrate that pelvic radiographs are valid inputs and to investigate the impact of multimodal integration and missing data handling.
> Training the CNN in Kong et al. end-to-end with a DeepSurv loss would be an interesting experiment for future work to further support performance claims.

---

> ### Comment · Reviewer_QArX · 2026-01-28
> **Comments on first responses**
>
> The authors improved the clarity of the manuscript greatly by adding detail. For me, some methodological points remain unclear which I will address below:
>
> - Kong et al. -- original vs. re-implementation and ambuguity of terminology: The authors describe their re-implementation in detail in 3.2, saying the feature extractor weights remained frozen. Also, in a comment, the authors say, the Kong et al. model is a “single-stage approach, as the pretrained ResNet is used as a feature extractor and the entire model is optimized jointly with a survival loss.” but later say it is not trained end-to-end, which, to me, is usually synonymous for “single-stage”. I read Kong et al. as most likely being a two-stage approach, glueing exported image-extracted features to clinical metadata and using this as the input to an out-of-the-box tabular deep CoxPH model, which is trained in isolation. The authors of the present work, on the other hand, did not freeze the features, but the encoder weights, and since augmentations were applied, one image could yield different feature vectors during training. This is absolutely valid, and it could be argued that this is neither an end-to-end nor a two-stage approach. Since this most likely deviates from the Kong et al. paper, the authors should differentiate between the original Kong et al. paper vs. their implementation: Referencing the Kong model as single-stage can be confusing, although valid for the version that was implemented, depending on the definition used for “single-stage”. A bit more context on this matter should be added to the baselines and/or related works section to avoid confusion.
>
> - Hyperparameter settings for dl baseline's DCTM head and dl baseline's image preprocessing: Sec. 3.2 states that "the same optimization strategy and training protocol" were used for dl baseline and proposed method. However, without further clarification, this is still ambiguous: Does this include (a) the same hyperparameter search, or (b) the same final hyperparameter settings as determined by the hyperparameter search from the proposed model, or (c) have entirely diffent settings been used for the dl baseline head? Does the quote include that all the image preprocessing was the same? If a direct reference to Sec. 3.4 was added to the above quote and Sec. 3.4 had a note on which hyperparameters the dl baseline's DCTM head used, this would be solved, in my opinion. Given that one can read the manuscript in a way that one does not know how the DCTM head settings differ between dl baseline and the proposed model (and DCTM has many settings and they likely influce the performance), a clarification is needed.
>
> - Tabular data preprocessing in Sec. 3.4: "categorical CRFs were one-hot encoded for the ATSF model, with variable names being encoded for clear identification. For the DL-baseline, categorical variables were included as binary indicators (either 0 or 1)" Does this mean that the authors collapsed some categorical variables with >2 levels to binary variables only for the DL baseline, while the proposed model received one-hot-encodings? Or were there no variables with >2 categories, and the descriptions are equivalent? Similarly, why does the proposed method receive a different preprocessing of continuous variables ("standardized" vs. "normalized to the [0,1] range")? It is not clear if this is done on purpose, e.g. if Kong et al. use this and the authors wanted to stick to their protocol, or, if it represents a limitation prohibiting too strong attribution claims regarding the proposed fusion approach.
>
> - Statistical test. I thank the authors for the clarification via commenting, but I think that the manuscript should include the information that only one test between the best own model and the best baseline was performed, too.

---

> > ### Author Response · Authors · 2026-01-29
> > **Response to Reviewer QArX (1/3)**
> >
> > ### Kong et al. -- original vs. re-implementation and ambuguity of terminology: The authors describe their re-implementation in detail in 3.2, saying the feature extractor weights remained frozen. Also, in a comment, the authors say, the Kong et al. model is a “single-stage approach, as the pretrained ResNet is used as a feature extractor and the entire model is optimized jointly with a survival loss.” but later say it is not trained end-to-end, which, to me, is usually synonymous for “single-stage”. I read Kong et al. as most likely being a two-stage approach, glueing exported image-extracted features to clinical metadata and using this as the input to an out-of-the-box tabular deep CoxPH model, which is trained in isolation. The authors of the present work, on the other hand, did not freeze the features, but the encoder weights, and since augmentations were applied, one image could yield different feature vectors during training. This is absolutely valid, and it could be argued that this is neither an end-to-end nor a two-stage approach. Since this most likely deviates from the Kong et al. paper, the authors should differentiate between the original Kong et al. paper vs. their implementation: Referencing the Kong model as single-stage can be confusing, although valid for the version that was implemented, depending on the definition used for “single-stage”. A bit more context on this matter should be added to the baselines and/or related works section to avoid confusion.
> >
> > - Thank you for highlighting the differences between the original work by Kong et al. and our re-implementation. We agree that the terminology is ambiguous.
> > Kong et al. (2022) describe a two-stage pipeline. In the first stage, a keypoint detection model is trained to localize the vertebrae L1–L5. Based on these keypoints, identical ROIs or image patches are cropped from all radiographs. These ROIs are fixed and are not modified further in the second stage. In the second stage, the ROIs, together with clinical risk factors, are used as input to a DeepSurv model, which is trained using the negative log likelihood loss. However, no backpropagation for image localization or ROI generation of the first stage is performed.
> > The paper by Kong et al. does not explicitly specify whether the weights of the CNN-based image encoder are updated during survival training or whether the encoder is effectively used as a frozen, pretrained feature extractor. Therefore, we had to make assumptions for  our re-implementation: Specifically, we considered only the second stage. The weights of the image encoder remain frozen and are not optimized with the DeepSurv loss. Accordingly, there is no end-to-end training from raw images to the survival output. We agree that referring to “end-to-end” in our earlier response is therefore misleading.
> > When considered outside the context of Kong et al., our implementation can be regarded as single-stage, since image features and clinical variables are optimized jointly within a single training process using one loss. Due to the use of data augmentation, multiple feature representations can be generated from the same image during training, even though the encoder weights remain fixed. This training strategy therefore differs both from a classical two-stage approach with fully pre-extracted features and from a strictly end-to-end trained model.
> > We agree with you that referring to the Kong model as “single-stage” without further clarification can be confusing, as this strongly depends on the underlying definition. To avoid this confusion, we now more carefully distinguish between the original approach by Kong et al. and our re-implementation in the revised manuscript. Furthermore, we explicitly clarify the underlying assumptions and terminology in the baselines and related work section.

---

> > > ### Author Response · Authors · 2026-01-29
> > > **Response to Reviewer QArX (2/3)**
> > >
> > > ### Hyperparameter settings for dl baseline's DCTM head and dl baseline's image preprocessing: Sec. 3.2 states that "the same optimization strategy and training protocol" were used for dl baseline and proposed method. However, without further clarification, this is still ambiguous: Does this include (a) the same hyperparameter search, or (b) the same final hyperparameter settings as determined by the hyperparameter search from the proposed model, or (c) have entirely diffent settings been used for the dl baseline head? Does the quote include that all the image preprocessing was the same? If a direct reference to Sec. 3.4 was added to the above quote and Sec. 3.4 had a note on which hyperparameters the dl baseline's DCTM head used, this would be solved, in my opinion. Given that one can read the manuscript in a way that one does not know how the DCTM head settings differ between dl baseline and the proposed model (and DCTM has many settings and they likely influce the performance), a clarification is needed.
> > >
> > > -Thank you for raising this point. Please find more details to the ambiguous statements below. We also added these clarification to the draft of our paper:
> > >
> > > **(a) Hyperparameter optimization.**
> > > For the DL baseline, no automatic hyperparameter optimization (HPO) was conducted. Instead, the hyperparameters were selected manually. This differs from our ATSF model.
> > >
> > > **(b) Final hyperparameter settings.**
> > > By “same optimization strategy and training protocol,” we refer to the use of the same optimizer, loss functions, and overall training procedure. This does not imply that the final hyperparameter values identified for the proposed method were reused for the DL baseline. We provide a table in the appendix for the exact hyperparameter that were used for the DL-baseline models.
> > >
> > > **(c) DCTM head settings for the DL baseline.**
> > > The DCTM head of the DL baseline uses its own explicitly defined set of hyperparameters. To avoid any remaining ambiguity, we added a direct reference from Sec. 3.2 to Sec. 3.4, where the DCTM head parameters used for the DL baseline are now explicitly stated.
> > > Image preprocessing.
> > >
> > > The image preprocessing pipeline is identical for the DL baseline and ATSF. Both methods receive the same preprocessed image data, ensuring a fair comparison.

---

> > ### Author Response · Authors · 2026-01-29
> > **Response to Reviewer QArX (3/3)**
> >
> > ### Tabular data preprocessing in Sec. 3.4: "categorical CRFs were one-hot encoded for the ATSF model, with variable names being encoded for clear identification. For the DL-baseline, categorical variables were included as binary indicators (either 0 or 1)" Does this mean that the authors collapsed some categorical variables with >2 levels to binary variables only for the DL baseline, while the proposed model received one-hot-encodings? Or were there no variables with >2 categories, and the descriptions are equivalent? Similarly, why does the proposed method receive a different preprocessing of continuous variables ("standardized" vs. "normalized to the [0,1] range")? It is not clear if this is done on purpose, e.g. if Kong et al. use this and the authors wanted to stick to their protocol, or, if it represents a limitation prohibiting too strong attribution claims regarding the proposed fusion approach.
> >
> > Thank you for pointing this out. We would like to clarify both aspects.
> >
> > All categorical CRFs in our dataset were binary. Therefore, one-hot encoding for the ATSF model and binary (0/1) indicators for the DL baseline are equivalent representations, and no multi-level categorical variables were collapsed. Both models received the same information content.
> >
> > Regarding continuous variables, they were standardized in both cases. The differing terminology (“standardized” vs. “normalized to the [0,1] range”) was imprecise in the original manuscript and has been corrected in the revised version. There are no difference in the preprocessing, and both models were trained on identically standardized continuous inputs.
> >
> > Overall, the preprocessing choices do not introduce an asymmetry between the proposed method and the baseline, nor do they limit the attribution of performance differences to the proposed fusion approach.
> >
> > ### Statistical test. I thank the authors for the clarification via commenting, but I think that the manuscript should include the information that only one test between the best own model and the best baseline was performed, too.
> >
> > - We have added this information to the revised manuscript, explicitly stating that the statistical test was conducted only between the best-performing proposed model and the best-performing baseline.

---

> > > ### Comment · Reviewer_QArX · 2026-01-30
> > >
> > > I thank the authors for all clarifications.
> > > Regarding the hyperparameters used for the dl baseline: I understand that the settings were selected manually; it, however, remains unknown to me, how. I appreciate that the settings will be added to the appendix and would suggest to further add a note on the "how": According to which premise have they been selected manually (default values vs. popular values vs. reasonable values vs. average/mode of main model's settings, etc.)? This would make explicit that strong, plausible settings have been used, and would provide justification for the baseline. I think that being explicit here could easily resolve any doubts; while guessed settings diverging strongly from the ATFS models would require a clear limitation note in the manuscript.

---

> > > > ### Author Response · Authors · 2026-01-30
> > > > **Response to Reviewer QArX**
> > > >
> > > > Thank you for the follow-up question. We iteratively adjusted hyperparameters such as the learning rate, data augmentation settings, and the number of Bernstein polynomials. For each configuration, we inspected the training and validation loss curves to identify if the DL-baseline was overfitting or underfitting. Based on this process, we obtained a final set of hyperparameters, from which we selected the configuration that achieved the best performance measured by the validation C-index while showing stable training behavior.
> > > >
> > > > In addition the tested hyperparameters came from the same parameter space as those used for the ATSF models, ensuring that the DL-baseline was evaluated with plausible and comparable settings.
> > > >
> > > > We did not apply a automated hyperparameter optimization for the DL-baseline, as full optimization was not the objective. We agree that the use of a manual hyperparameter selection is a limitation, which we will add then in the final manuscript.

---

### Author Rebuttal · Authors · 2026-01-23

**Rebuttal:**

We would like to thank all reviewers for their thorough and constructive comments on our manuscript. We welcome the overall positive sentiment and appreciate that our method is seen to have a clear clinical motivation and relevance [$\textcolor{red}{\text{QArX}}$, $\textcolor{blue}{\text{6gRv}}$], that our method is able to generate performance gains, and that our results and conclusions were considered convincing and robust [$\textcolor{red}{\text{QArX}}$, $\textcolor{blue}{\text{6gRv}}$, $\textcolor{green}{\text{djke}}$]. We also appreciate that the broad applicability of the approach, in particular the use of widely available radiological image data and the robust handling of missing clinical variables, was also positively highlighted [$\textcolor{green}{\text{djke}}$]. Finally, the methodological foundation [$\textcolor{red}{\text{QArX}}$, $\textcolor{blue}{\text{6gRv}}$], the statistical evaluation including ablation studies [$\textcolor{red}{\text{QArX}}$], the interpretable attention mechanisms [$\textcolor{green}{\text{djke}}$], and the the underlying dataset were also perceived as strengths of the work [$\textcolor{blue}{\text{6gRv}}$].

The reviewers’ feedback has helped to greatly improve the exposition of our revised manuscript, which we uploaded along with our responses to the detailed comments. We added relevant technical details, clearer justifications for individual design decisions, and a more precise presentation of related work and the DL-baseline. In addition, we refined the discussion on aspects such as the time horizon and the validity of the employed metrics [$\textcolor{red}{\text{QArX}}$, $\textcolor{blue}{\text{6gRv}}$]. Further ablation analyses and the possible consideration of BMD or the course of treatment have given us valuable impulses for the revision [$\textcolor{green}{\text{djke}}$].

**Supporting Material:**

/attachment/e5384c68ac09a2d74d1859ce4981855c70a42279.pdf

---

### Meta-Review · Area_Chair_z1z9 · 2026-02-09

**Recommendation:** Accept (Poster)
**Confidence:** 4

**Metareview:**

All reviewers provide positive feedback (one of them raised score after the rebuttal).  Overall, this is solid work with relevant contributions, although the experimental scope and methodology depth can be further improved.

---

### Decision · Program_Chairs · 2026-02-13

Accept (Poster)